# Hyaluronic Acid-Conjugated with Hyperbranched Chlorin e6 Using Disulfide Linkage and Its Nanophotosensitizer for Enhanced Photodynamic Therapy of Cancer Cells

**DOI:** 10.3390/ma12193080

**Published:** 2019-09-21

**Authors:** Shin Jung, Seunggon Jung, Doo Man Kim, Sa-Hoe Lim, Yong Ho Shim, Hanjin Kwon, Do Hoon Kim, Chang-Min Lee, Byung Hoon Kim, Young-Il Jeong

**Affiliations:** 1Department of Neurosurgery, Chonnam National University Hwasun Hospital, Hwasun 58128, Korea; sjung@chonnam.ac.kr (S.J.); sahoe@cnuh.com (S.-H.L.); 2Brain Tumor Research Laboratory, Chonnam National University Research Institute of Medical Sciences, Chonnam National University Hwasun Hospital, Hwasun 58128, Korea; 3Department of Oral and Maxillofacial Surgery, School of Dentistry, Chonnam National University, Gwangju 61186, Korea; seunggon.jung@chonnam.ac.kr; 4Department of Materials Science and Engineering, Chonnam National University, Gwangju 61186, Korea; kmdooman@nate.com; 5UltraV Co. Ltd. R&D Center, Seoul 04779, Korea; rodysss@naver.com (Y.H.S.); kwonhanji@gmail.com (H.K.); 6Department of Integrative Physiology and Pathobiology, Tufts University School of Medicine, Boston, MA 02111, USA; DoHoon.Kim@tufts.edu; 7Department of Dental Materials, College of Dentistry, Chosun University, Gwangju 61452, Korea; ckdals1924@daum.net; 8Research Institute of Convergence of Biomedical Sciences, Pusan National University Yangsan Hospital, Gyeongnam 50612, Korea

**Keywords:** Chlorin e6, branched polymer, redox sensitive, CD44-receptor, photodynamic therapy

## Abstract

The main purpose of this study is to synthesize novel types of nanophotosensitizers that are based on hyperbranched chlorin e6 (Ce6) via disulfide linkages. Moreover, hyperbranched Ce6 was conjugated with hyaluronic acid (HA) for CD44-receptor mediated delivery and redox-sensitive photodynamic therapy (PDT) against cancer cells. Hyperbranched Ce6 was considered to make novel types of macromolecular photosensitizer since most of the previous studies regarding nanophotosensizers are concerned with simple conjugation between monomeric units of photosensitizer and polymer materials. Hyperbranched Ce6 was synthesized by conjugation of Ce6 each other while using disulfide linkage. To synthesize Ce6 tetramer, carboxyl groups of Ce6 were conjugated with cystamine and three equivalents of Ce6 were then conjugated again with the end of amine groups of Ce6-cystamine. To synthesize Ce6 decamer as a hyperbranched Ce6, six equivalents of Ce6 was conjugated with the end of Ce6 tetramer via cystamine linkage. Furthermore, HA-cystamine was attached with Ce6 tetramer or Ce6 decamer to synthesize HA-Ce6 tetramer (Ce6tetraHA) or HA-Ce6 decamer (Ce6decaHA) conjugates. Ce6tetraHA and Ce6decaHA nanophotosensitizers showed small diameters of less than 200 nm. The addition of dithiothreitol (DTT) and hyaluronidase (HAse) induced a faster Ce6 release rate in vitro drug release study, which indicated that Ce6tetraHA nanophotosensitizers possess redox-sensitive and HAse-sensitive release properties. Ce6tetraHA nanophotosensitizers showed higher intracellular Ce6 accumulation, higher ROS generation, and higher PDT efficacy than that of Ce6 alone. Ce6tetraHA nanophotosensitizers responded to the CD44 receptor of cancer cell surface, i.e., the pre-treatment of HA blocked CD44 receptor of U87MG or HCT116 cells and then inhibited delivery of nanophotosensitizers in vitro cell culture study. Furthermore, *in vivo* tumorxenograft study showed that fluorescence intensity in the tumor tissues was stronger than those of other organs, while CD44 receptor blocking by HA pretreatment induced a decrease of fluorescence intensity in tumor tissues when compared to liver. These results indicated that Ce6tetraHA nanophotosensitizers delivered to tumors by redox-sensitive and CD44-sensitive manner.

## 1. Introduction

Photodynamic therapy (PDT) is generally consisted of light, photosensitizer, and oxygen [1]. PDT is believed to be a safe therapeutic regimen for cancer patients, since it has little undesirable side-effects when compared to traditional chemotherapy [1,2,3]. Photosensitizer produces reactive oxygen species (ROS) under light irradiation at the desired site of action [3,4,5]. When tumor tissues are irradiated with appropriate wavelength of light, photosensitizers produce ROS in the irradiated site only, but not in the normal counterparts [5,6,7]. Subsequently, photosensitizers kill the cancer cells by over-production of ROS in tumor tissues. Ce6 has been extensively investigated for PDT of cancer, because it has deeper tissue penetration efficiency and it is activated at higher wavelength of light than traditional photosensitizers, such as 5-aminolevulic acid (5-ALA) or porphyrins, [8,9]. Additionally, higher wavelength of light can penetrate tissues deeper than lower wavelength of light [6,7,8,9]. Despite of these advantages, most of photosensitizers, including Ce6, still have low cancer cell specificity and low delivery capacity against the tumor tissues and requirement for long-term sun-shade [3,9,10,11]. These drawbacks still limit the clinical application of photosensitizers [2,12].

Many scientists have developed nanophotosensitizers in last decades [13,14,15,16,17]. For example, Chin et al., reported that Ce6-polyvinylpyrrolidone (PVP) formulations revealed a faster clearance rate from the body when compared to Ce6 alone [13]. These properties of Ce6-PVP formulations eliminate the requirement for long-period photosensitivity precautions. We previously reported that nanophotosensitizers that are composed of poly(ethylene glycol) (PEG)-Ce6 conjugates showed superior aqueous solubility, higher fluorescence intensity in aqueous solution, and higher PDT efficacy against HCT116 colon carcinoma cells *in vitro* and *in vivo* as compared to Ce6 alone [15]. PEG-Ce6 nanophotosensitizer revealed enhanced penetration efficiency in the colonic region and CT26 tumor tissues when compared to Ce6 alone. Furthermore, the nanophotosensitizer of chitosan-Ce6 complexes has higher PDT efficacy against cholangiocarcinoma cells and higher absorption efficiency in bile duct tissue explants when compared to Ce6 alone [18]. Nanomedicine-based PDT also enables disease diagnosis and image-guided surgery [19,20,21]. Muhanna et al., reported that multimodal porphyrin lipoprotein-mimicking nanoparticle (PLPs) can be applied in fluorescence imaging, positron emission tomography (PET), and PDT [20]. They argued that PLPs easily detect primary tumor and metastatic nodes [20]. Furthermore, photosensitizer-conjugated HA can be used to diagnosis and therapeutic purposes [22,23]. Yoon et al., reported that Ce6-conjugated HA nanoparticles can be delivered tumor-specific enzyme, such as HAse and CD44-receptor-mediated pathway [22]. Gao et al., reported that the Ce6-encapsulated HA nanoparticles were efficiently accumulated in the tumor xenograft of human colon cancer and observed no apparent systemic toxicity against mice [23]. 

The biochemical and pathophysiological characters of microenvironment of tumor tissues are quite different to normal counterpart [24]. The tumor cells are characterized by a higher expression of receptors, higher enzymatic activity, such as hyaluronidase (HAse), poor perfusion, acidic pH environment, and higher reduction/oxidation (redox) status when compared to normal cells [24,25,26]. Subsequently, the tumor microenvironment has quite different physiological states as compared to normal tissues. Furthermore, aggressive malignant cells highly express CD44 or RHAMM, which is known as hyaluronic acid (HA) receptor [27]. CD44 receptor is related to the growth, migration, and metastasis of cancer cells [27]. Many scientists have investigated HA as a targeting moiety of nanomedicine for long decades [26,28,29,30]. Lee and Jeong reported that the nanoparticles of HA-Ce6-poly(L-histidine) conjugates specifically deliver the anticancer drug to a CD44 receptor of MDA-MB 231 cells at *in vitro* and *in vivo* [30]. They argued that Ce6-conjugated HA nanoparticles can be used as a theranostic platform of anticancer drug delivery. On the other hand, abnormal redox status of tumor microenvironment is also spotlighted in recent decades, since high metabolic activity in tumor tissues promotes redox activities [28,29,30,31,32]. The abnormal redox status of tumor has been investigated by many scientists as a target for stimuli-sensitive delivery of bioactive agents [28,29,30,31,32,33]. Since the intracellular glutathione (GSH) level in cancer cells is normally higher than that in normal cells, these properties can be applied for cancer targeting. Since the disulfide bond between drug and vehicle can be disintegrated by GSH, nanoparticles can be designed to respond intracellular GSH content in cancer cells [28,29,30,31,32,33].

For present study, we designed nanophotosensitizer while using hyperbranched Ce6-conjugated HA. Hyperbranched Ce6 was synthesized by the conjugation of Ce6 each other using disulfide linkage. Furthermore, Ce6tetraHA or Ce6decaHA conjugates must be formed nanoscale vehicles as nanophotosensitizers since Ce6 itself is a hydrophobic molecule, while HA is a hydrophilic macromolecule. Therefore, hyperbranched nanophotosensitizers can be used as therapeutic platform for PDT of cancer. As hyperbranched Ce6 and HA is connected by disulfide linkages, Ce6tetraHA or Ce6decaHA conjugates should be degraded in tumor cells by glutathione (GSH) and HA itself can be degradable by HAse. We characterized the biochemical and physicochemical properties of Ce6tetHA nanophotosensitizer *in vitro* and *in vivo*.

## 2. Materials and Methods

### 2.1. Materials

HA (molecular weight: 7,460 g/mol) and Ce6 were purchased from Lifecore Biomedical (Chaska, MN, USA) and Frontier Sci. Co. (Logan, UT, USA), respectively. Hyaluronidase (HAse) from bovine tastes, Cystamine, L-glutathione (GSH), 1,4-dithiothreitol (DTT), N-hydroxysuccinimide (NHS), N-(3-dimethylaminopropyl)-N’-ethylcarbodiimide hydrochloride (EDAC), triethylamine (TEA), 3-(4,5-Dimethyl-2-thiazolyl)-2, 5-diphenyl-2H-tetrazolium bromide (MTT), 2′,7′-dichlorofluorescin diacetate (DCFH-DA), dimethylsulfoxide (DMSO), cremophor^®^ EL, and sodium cyanoborohydride were purchased from Sigma Chem. Co. (St. Louis, MO, USA). The dialysis membranes having molecular weight cutoffs (MWCO) of 1000, 2000, and 8000 g/mol were purchased from Spectra/Por^TM^ Membranes (Spectrum® Chem. MFG Co., New Brunswick, NJ, USA). Organic solvents were used as HPLC or extra-pure grade.

### 2.2. Synthesis of Ce6tetraHA Conjugates

Ce6 tetramer: Ce6 (0.1 mM, 59.7 mg) dissolved in DMSO (10 mL) was mixed with three equivalent amounts of EDAC (0.3 mM, 57.51 mg) and NHS (0.3 mM, 34.53 mg). This solution was magnetically stirred for 6h. Following this, fifteen equivalents of cystamine HCl (1.5 mM, 337.8 mg) was dissolved in 10 mL water/DMSO mixtures (2/8, v/v) and then added to Ce6 solution with trace amount of TEA following with magnetic stirring over 36 h. The resulting solution was introduced into dialysis membrane (MWCO: 1000 g/mol) and then dialyzed against distilled water for two days. Distilled water was exchanged every 3 h intervals to avoid saturation. Dialyzed solution was freeze-dried for three days and Ce6 functionalized with three cystamine moieties was obtained as a solid (Ce6-(-NSSNH_2_)_3_). After that, Ce6 (0.3 mM, 179.1 mg) dissolved in 10 mL DMSO was mixed with an equivalent amount of EDAC (0.3 mM, 57.51 mg) and NHS (0.3 mM, 34.53 mg). This solution was then stirred for 9 h and then mixed with Ce6-(-NSSNH_2_)_3_ (98 mg). The resulting solution was further stirred for two days and then introduced into dialysis membrane (MWCO: 1000 g/mol). Subsequently, this was dialyzed against distilled water for two days with exchange of water every 3 h intervals to remove unreacted chemicals and by-products. This solution was lyophilized for 3 days to obtain Ce6 tetramer as a solid. The final yield of Ce6 tetramer was about 96% (w/w). Yield = [Ce6 tetramer weight/(Ce6 weight + Ce6-(-NSSNH_2_)_3_ weight)] × 100.

Ce6 decamer: Ce6 tetramer (274 mg) dissolved in 20 mL DMSO was mixed with six equivalent amounts of EDAC (0.6 mM, 115.02 mg) and NHS (0.6 mM, 69.05 mg). This was stirred for 12 h and then ten equivalents of cystamine HCl (6 mM, 1351 mg) in 10 mL DMSO/water (8/2, v/v) mixtures was added. This was stirred for 36 h to be reacted. The resulting solution was introduced into dialysis membrane (MWCO: 2000 g/mol) and then dialyzed against distilled water for two days. Following this, the resulting solution was then lyophilized for three days to obtain Ce6 tetramer functionalized with six cystamine moieties as a sold. Ce6 (0.6 mM, 358.2 mg) dissolved in 10 mL DMSO was mixed with equivalent amounts of EDAC (0.6 mM, 115.02 mg) and NHS (0.6 mM, 69.06 mg) (Ce6-NHS). This solution was then magnetically stirred for 6 h. Ce6-NHS solution was mixed with Ce6 tetramer-cystamine (350 mg in 15 mL DMSO) and then stirred for 36 h. This was dialyzed against distilled water while using dialysis membrane (MWCO: 2000 g/mol) for two days following lyophilization for three days. Afterwards, Ce6 decamer was obtained as a solid. The yield of Ce6 decamer was higher than 97% (w/w). Yield = [Ce6 decamer weight/(Ce6 tetramer weight + Ce6 weight)] × 100.

HA-cystamine: HA (160 mg) dissolved in 5 mL of water/DMSO mixtures (1/4, v/v) was mixed with excess quantity of sodium cyanoborohydride (31 mg) and then magnetically stirred for 12 h. After that, this solution was precipitated into an excess quantity of methanol and collected by filtration with filter paper. The precipitates were washed three times with methanol and dried in vacuum oven for two days at room temperature. HA with reductive end (80 mg) dissolved in 10 mL water/DMSO (2/8, v/v) and then ten equivalents of cystamine (22.5 mg, 0.1 mM) was added with trace amounts of TEA. This solution was stirred at room temperature for 36 h. After that, the resulting solution was introduced into dialysis tube (MWCO: 2000 g/mol) and dialyzed against water for two days. This was lyophilized for two days to obtain HA-cystamine as a solid.

Ce6tetraHA conjugates: 27.5 mg of Ce6 tetramer was dissolved in 10 mL DMSO with EDAC (1.91 mg, 0.01 mM) and NHS (1.15 mg, 0.01 mM). This solution was magnetically stirred for 9 h and then mixed with 81 mg HA-cystamine conjugates in 5 mL water/DMSO mixtures (1/4, v/v). This solution was stirred for 36 h and then put into dialysis membrane (MWCO: 8000 g/mol) to dialyze against water over two days. Following this, the resulting solution was lyophilized for three days to obtain Ce6tetraHA conjugates as a solid. This solid was stored in refrigerator at −20 °C. Yield was approximately 97% (w/w) from mass measurement: Yield = [Weight of final product/(Weight of Ce6 tetramer + weight of HA-cystamine conjugates)] × 100.

Ce6decaHA conjugates: Ce6 decamer (70 mg) was dissolved in 15 mL DMSO with EDAC (1.91 mg, 0.01 mM) and NHS (1.15 mg, 0.01 mM). This solution was stirred magnetically for 9 h and then mixed with 81 mg HA-cystamine conjugates in 5 mL water/DMSO mixtures (1/4, v/v). This solution was further stirred for 36 h and then put into dialysis membrane (MWCO: 8000 g/mol). This was dialyzed against water over two days and lyophilized for three days to obtain Ce6decaHA conjugates as a solid. The final product was used to analyse or store in refrigerator at −20 °C. Yield was approximately 98% (w/w) from mass measurement: Yield = [Weight of final product/(Weight of Ce6 decamer + weight of HA-cystamine)] × 100. 

### 2.3. Preparation of Ce6tetraHA/Ce6decaHA Nanophotosensitizers

Ce6tetraHA or Ce6decaHA conjugates (20 mg) was dissolved in 5 mL water/DMSO mixtures (1/4, v/v) and then introduced into dialysis membrane (MWCO: 8000 g/mol). These were dialyzed against distilled water for one day with exchange of water every 2~3 h intervals. The resulting solution was used for analysis or drug release study.

Ce6 contents: Ce6tetraHA or Ce6decaHA nanophotosensitizers (5 mg) were reconstituted in 50 mL PBS in the presence of 20mM DTT. Subsequently, this was stirred for 24 h. One mL of this solution was diluted ten times with DMSO and then Ce6 concentration was evaluated with an Infinite M200 pro microplate reader (Tecan, Mannedorf, Switzerland) at excitation wavelength of 407 nm and emission wavelength of 664 nm. For the standard test, Ce6 alone in DMSO was diluted 20 times with PBS (20 mM DTT) and then diluted ten times with DMSO.

Ce6 content (wt %) = (Ce6 weight/total weight of nanophotosensitizers) × 100.

Ce6 contents in Ce6tetraHA nanophotosensitizers and Ce6decaHA nanophotosensitizers were approximately 22.1% (w/w) and 39.8% (w/w), respectively.

### 2.4. Characterization of Ce6tetraHA or Ce6decaHA Conjugates and Nanophotosensitizers

The synthesis of Ce6tetraHA/Ce6decaHA conjugates was confirmed with ^1^H NMR spectra (500 mHz superconducting Fourier transform (FT)-NMR spectrometer, Varian Unity Inova 500 MHz NB High Resolution FT NMR; Varian Inc., Santa Clara, CA, USA).

The particle size of Ce6tetraHA or Ce6decaHA nanophotosensitizers (concentration: 0.1%, w/w) was analyzed with Zetasizer Nano-ZS (Malvern, Worcestershire, UK). To estimate average particle size, they were measured particle sizes at least three times and expressed as average ± S.D.

The morphology of Ce6tetraHA nanophotosensitizers was observed with transmission electron microscope (TEM) (H-7600, Hitachi Instruments Ltd., Tokyo, Japan). One drop of nanophotosensitizer solution was put onto the carbon film coated grid. Subsequently, this was dried in room temperature for 3 h, followed by negative staining with phosphotungstic acid (0.1%, w/w in deionized water). The observation of nanophotosensitizer morphology was performed at 80 kV.

### 2.5. Fluorescence Emission Scan of Nanophotosensitizers

The fluorescence properties of nanophotosensitizers were performed with emission scan while using Infinite M200 pro microplate reader between 500 nm and 800 nm (excitation wavelength: 400 nm). Fluorescence images of same solution were scanned with Maestro 2 small animal imaging instrument (Cambridge Research and Instrumentation Inc., MA, USA). Ce6tetraHA nanophotosensitizers prepared described above were reconstituted in PBS (pH 7.4, 0.01 M) in the presence of various concentrations of DTT or HAse and then incubated them at 37 °C for 3 h. These were used to analyze the fluorescence properties.

### 2.6. Ce6 Release from Nanophotosensitizer

Ce6 release from Ce6tetraHA or Ce6decaHA nanophotosensitizers was performed in PBS (0.01 M, pH 7.4) in the presence of DTT. The volume of nanophotosensitizer solution was adjusted to 20 mL with deionized water and then 5 mL was introduced into dialysis membrane (MWCO = 8000 g/mol). Dialysis membrane was introduced into 50 mL conical tube with 35 mL PBS (pH 7.4, 0.01 M) in the presence of various concentrations of GSH or HAse. Subsequently, these were incubated in 37 °C under shaking at 100 rpm. At specific time intervals, the whole PBS was taken to evaluate liberated Ce6 in PBS. Following this, fresh PBS was introduced. The liberated concentration of Ce6 in PBS was evaluated with an Infinite M200 pro microplate reader (Tecan) (excitation wavelength: 407 nm, emission wavelength: 664 nm). All of the experiments were triplicated and the results were expressed as mean ± standard deviation (S.D.).

### 2.7. Cell Culture

NIH3T3 mouse fibroblast cells, U87MG human malignant glioma cells, and HCT116 mouse colon carcinoma cells were obtained from Korean Cell Line Bank (Seoul, Korea). NIH3T3 cells and U87MG cells were cultured in DMEM (Gibco, Grand Island, NY, USA) supplemented with 10% (v/v) heat-inactivated fetal bovine serum (FBS) (Invitrogen) and 1% (v/v) penicillin-streptomycin at 37 °C in a 5% CO_2_ incubator. The HCT116 cells were cultured in RPMI1640 (Gibco, Grand Island, NY, USA) supplemented with 10% (v/v) heat-inactivated fetal bovine serum (FBS) (Invitrogen) and 1% (v/v) penicillin-streptomycin.

### 2.8. PDT Treatment

U87MG cells or HCT116 cells (2 × 10^4^ cells) were seeded in 96 well plates. For Ce6 treatment, Ce6 was dissolved in DMSO and then diluted with media at least one hundred times (Final concentration of DMSO was less than 0.5% (v/v)). For the treatment of hyperbranched Ce6, Ce6 tetramer, or Ce6 decamer dissolved in DMSO was diluted with media (Final DMSO concentration was less than 0.5% (v/v)). For the treatment of nanophotosensitizers, Ce6tetraHA or Ce6decaHA nanophotosensitizers prepared, as described above, were filtered with 1.2 μm syringe filter and then diluted with media. Each treatment was incubated for 2 h in CO_2_ incubator (37 °C). Subsequently, the cells were washed with PBS and 100 μL serum-free fresh media was added. The cells were exposed to 664 nm light while using expanded homogenous beam (SH Systems, Gwangju, Korea). The dose of light irradiation was 2.0 J/cm^2^. The light intensity was measured with a photo radiometer (Delta Ohm, Padua, Italy) and cells were irradiated for 9 min. 29 s for 2.0 J/cm^2^ in our system. Irradiated cells were incubated for 24 h in CO_2_ incubator at 37 °C. Cell viability was performed with MTT proliferation assay, as follows: 30 μL MTT solution (5 mg/mL in PBS) was added and incubated for 4 h in CO_2_ incubator. Supernatants were discarded and replaced with 100 μL DMSO. Cell viability was evaluated with absorbance (570 nm) while using an Infinite M200 pro microplate reader (Tecan, Mannedorf, Switzerland). All procedures were performed in a dark condition. All of the values are average ± S.D. (Standard deviation) from eight wells.

Intrinsic toxicity or dark toxicity of nanophotosensitizers was carried out in thedark condition without light irradiation of normal cells or cancer cells.

### 2.9. Intracellular Ce6 Uptake of Nanophotosensitizers

Cells (2 × 10^4^ cells) in a 96-well plate were treated with Ce6 alone or nanophotosensitizers. Ninety minutes later, the cells were washed with PBS twice and then solubilized with 50 μL of lysis buffer (GenDEPOT, Barker, TX, USA). Relative fluorescence intensity as a Ce6 uptake ratio was measured with Infinite M200 pro microplate reader (Tecan) (Excitation wavelength: 407 nm, emission wavelength: 664 nm). All of the values are average ± S.D. (Standard deviation) from eight wells.

### 2.10. Fluorescence Observation of Cells

Cancer cells (3 × 10^5^ cells) seeded on the cover glass in six-well plates were exposed to Ce6 or nanophotosenstitizer for 90 min. After that, the cells were washed with PBS twice and then fixed with immobilization solution (Immunomount, thermo Electron Co. Pittsburgh, PA, USA). The cells were observed with fluorescence microscope (Eclipes 80i; Nikon, Tokyo, Japan).

### 2.11. ROS Generation

ROS generation was estimated with DCFH-DA. Various concentrations of Ce6 or nanophotosensitizer in each RPMI media with DCFH-DA (final concentration: 20 µM) were treated to cells (2 × 10^4^ cells) for 2 h at 37 °C. Following this, the cells were washed with PBS twice and replaced with 100 µL fresh phenol red free RPMI media. The cells were irradiated at 664 nm (2.0 J/cm^2^). Intracellular ROS generation in cancer cells was evaluated with microplate reader (Infinite M200 pro microplate reader (Tecan), Excitation wavelength, 485 nm; emission wavelength, 535 nm). All of the values are the average ± S.D. (Standard deviation) from eight wells.

### 2.12. In vivo Fluorescence Imaging Study Using Animal Tumor Xenograft

U87MG cells or HCT116 cells (1 × 10^6^ cells) were subcutaneously injected into the left side of back of nude BALb/C mice (Male, 20 g, five weeks old). When the diameter of solid tumor became larger than 4 mm, Ce6tetraHA nanophotosensitizer solution (5.0 mg/kg) was intravenously administered (i.v.) via the tail vein of the mice. The volume of injection solution was 100 μL. For the blocking of CD44 receptor of cancer cells, HA (20 mg/kg) in PBS was i.v. injected 1 h before nanophotosensitizer injection. 24 h later, the mice were sacrificed and then observed with Maestro^TM^ 2 small animal imaging instrument.

All animal studies were carried out under the guidelines of the Pusan National University Institutional Animal Care and Use Committee (PNUIACUC). The protocol of animal study has been reviewed and strictly monitored by the PNUIACUC on their ethical procedures and scientific care, and has been approved (Approval Number: PNU-2017-1610).

### 2.13. Statistical Analysis

The statistical significance of the results was evaluated with Student’s t test while using SigmaPlot^®^ program.

## 3. Results

### 3.1. Synthesis of Ce6tetraHA or Ce6decaHA Conjugates

Since Ce6 has three carboxylic acids in its chemical structure, the hyper-branched Ce6 tetramer or decamer can be synthesized by conjugating each other while using cystamine, as shown in Figure 1. First of all, three carboxylic acid of Ce6 was activated with EDAC/NHS system to make Ce6-NHS. This was conjugated with cystamine to obtain cystamine-conjugated Ce6. Unreacted cystamine was removed by the dialysis procedure. As shown in Figure 1a, specific peaks of Ce6 were confirmed at 1.6–1.8 ppm (–CH_3_) and 6.0~7.0 ppm (–CH=CH_2_), while cystamine itself has specific peaks at 2.8~3.2 ppm (–CH_2_–CH_2_–) (Data not shown). Peaks of Ce6 and cystamine were confirmed at Ce6-cystamine, as shown in Figure 1a, which indicates that Ce6-cystamine have functional amine end group with disulfide. To the end of amine groups, three equivalents of NHS-activated Ce6 were conjugated again to produce Ce6 tetramer, as shown in Figure 1b and Appendix A. The yield of Ce6 tetramer was about 96%. As shown in Figure 1b, specific peaks of Ce6 and cystamine were confirmed at 1.0 ppm, 6.0~7.0 ppm, and 2.8~3.2 ppm, respectively. These results indicated that the Ce6 tetramer could be synthesized.

As six carboxylic acids of Ce6 tetramer are remained, the carboxylic acids of Ce6 tetramer was conjugated with cystamine and then six equivalents of Ce6 was conjugated once more to produce Ce6 decamer, as shown in Figure 1c and Appendix A. As shown in Figure 1c, peaks of Ce6 and cystamine was confirmed at 1.0 ppm, 6.0~7.0 ppm, and 2.8~3.2 ppm, respectively. These results indicated that the Ce6 decamer could be also synthesized. To estimate the number of Ce6 in Ce6 tetramer and Ce6 decamer, methoxy poly(ethylene glycol) (M.W. = 2000 g/mol) was conjugated with Ce6 tetramer of Ce6 decamer as shon in Appendix A. The estimated number of Ce6 of Ce6 tetramer and decamer was 3.64 and 9.1, respectively (Appendix A).

To the end of carboxyl group of Ce6 tetramer or Ce6 decamer, HA was further conjugated to make Ce6tetraHA or Ce6decaHA conjugates, as shown in Figure 2. Prior to synthesizing Ce6tetraHA or Ce6decaHA conjugates, HA-cystamine was prepared, as shown in Figure 2a. To the reduction end of HA, cystamine was conjugated to produce HA-cystamine (Figure 2a). The peaks of HA were confirmed at 3~4 ppm, while peaks of cystamine was appeared at 2.8 ppm, which indicated that cystamine was successfully conjugated with HA. HA-cystamine was attached to the carboxyl group of Ce6 tetramer to produce Ce6tetraHA conjugates (Figure 2b). As shown in Figure 2a,b, cystamine and Ce6 peaks were also confirmed, indicating that the Ce6tetraHA or Ce6decaHA conjugates were successfully synthesized.

Figure 3 showed the FT-IR measurements of hyperbranched Ce6. Carboxyl groups of Ce6 and primary amine groups of cystamine were confirmed at 1690 cm^−1^ and 1500~1650 cm^−1^, respectively. When cystamine was conjugated with Ce6, peaks of carboxylic acid of Ce6 were disappeared, while amine peaks (cystamine) were found at 1500~1650 cm^−1^. However, the carboxyl group of Ce6 was found at 1690 cm^−1^ when Ce6 was conjugated with Ce6-cystamine, as shown in Figure 3. Furthermore, HA was also confirmed at Ce6tetraHA conjugates, as shown in Figure 3. These results confirmed again synthesis of Ce6tetraHA conjugates, as shown in Figure 1 and Figure 2.

### 3.2. Preparation and Characterization of Ce6tetraHA Nanophotosensitizer

The Ce6tetraHA or Ce6decaHA conjugates were dialyzed against water to produce nanophotosensitizers. Nanophotosensitizers having core-shell structure can be formed in an aqueous environment, because HA and Ce6 have a hydrophilic and lipophilic properties, respectively. Hyperbranched Ce6 formed an inner-core of the nanophotosensitizers, while HA formed an outer-shell. Nanophotosensitizers of Ce6tetraHA and Ce6decaHA can be instantly reconstituted in the aqueous solution, such as deionized water or PBS. Figure 4 shows the morphological observation and particle size of Ce6tetraHA nanophotosensitizers. Ce6tetraHA nanophotosensitizers revealed the spherical shapes with a small diameter of less than 100 nm (Figure 4a). Furthermore, Figure 4b showed that the particle size of Ce6tetraHA nanophotosensitizers revealed a mono-dispersed pattern with small particle size distribution and diameter was less than 200 nm. As shown in Table 1, the average particle sizes of Ce6tetraHA or Ce6decaHA nanophotosensitizers were 82.6 ± 5.6 nm and 181.4 ± 12.3 nm from three different measurements, respectively. These results indicated that the Ce6tetraHA or Ce6decaHA conjugates successfully form small nanoparticles in aqueous solution. Since Ce6 in the Ce6tetraHA or Ce6decaHA nanophotosensitizers linked each other with disulfide linkage and disulfide group can be degraded by GSH or DTT, Ce6 can be specifically liberated by GSH from the nanophotosensitizers (Figure 4c). Ce6 itself showed similar fluorescence intensity in DMSO or PBS (Appendix A). Furthermore, Ce6, Ce6 tetramer and Ce6 decamer showed similar fluorescence pattern in DMSO as shown in Appendix A. Fluorescence spectra of Ce6tetraHA nanophotosensitizers was steadily increased according to DTT concentration, which indicated that Ce6 was released from nanophotosensitizers by the degradation of disulfide group. Furthermore, the effect of HAse addition in the nanophotosensitizer solution was also investigated, since HA can be degraded by HAse and then Ce6 or Ce6tetramer can be liberated. The fluorescence of nanophotosensitizer solution was increased by addition of 100 units of HAse and, especially, further increased when HAse and DTT were simultaneously added (Figure 4d). These results indicated that nanophotosensitizers respond to HAse and DTT. HA, an outershell of nanophotosensitizers, can be degraded by HAse and affected to Ce6 or Ce6 tetramer release.

Figure 5 showed the Ce6 release dynamics from the nanophotosensitizers. As shown in Figure 5a,b, the absence of DTT induced slow release rate of Ce6 from nanophotosenmsitizers. However, Ce6 release dynamics became significantly higher in the presence of DTT and release dynamics responded to the DTT concentration. Furthermore, the addition of HAse also increased the release dynamics of Ce6, as shown in Figure 5c. Especially, the dual addition of HAse and DTT also significantly increased the release dynamics of Ce6 from nanophotosensitizers, indicating that Ce6 release dynamics from nanophotosensitizers can be controlled by redox-sensitive manner and by HAse.

### 3.3. PDT Efficacy of Nanophotosensitizers at In Vitro Cell Culture

PDT efficacy of Ce6tetraHA or Ce6decaHA nanophotosensitizers was evaluated with U87MG human malignant glioma cells and HCT116 human colon carcinoma cells. Prior to PDT evaluation of nanophotosensitizers, intrinsic dark toxicity of Ce6 itself, and nanophotosensitizers with normal cells (NIH3T3) and cancer cells was evaluated, as shown in Figure 6. Ce6 and nanophotosensitizers both have little toxicity until 8.375 μM (5 μg/mL) against NIH3T3 cells and cancer cells in the dark condition. The viability of cells was maintained at higher than 80% until 8.375 μM (5 μg/mL) at all formulations, which indicates that Ce6tetraHA or Ce6decaHA nanophotosensitizers have low dark toxicity against normal cells and cancer cells as well as Ce6.

Figure 7 shows the Ce6 uptake ratio of Ce6 alone or nanophotosensitizers in cancer cells *in vitro*. Ce6 uptake ratio was dose dependently increased at all formulations (Figure 7a). Especially, Ce6 uptake ratio of nanophotosensitizers was significantly higher than Ce6 alone. Practically, no differences in Ce6 uptake ratio were found between the Ce6 tetramer, Ce6 decamer, Ce6tetraHA nanophotosensitizers, and Ce6decaHA nanophotosensitizers. Fluorescence images of cells also showed the superiority of nanophotosensitizers, i.e., red fluorescence color of nanophotosensitizers was stronger than that of Ce6 alone. Figure 8 showed the ROS generation (Figure 8a) and PDT efficacy (Figure 8b) against HCT116 cells and U87MG cells. ROS generation steadily increased according to the concentration of Ce6 (Figure 8a). Due to the higher uptake ratio, ROS generation of Ce6tetraHA and Ce6decaHA nanophotosensitizers was higher than that of Ce6 alone. Furthermore, the PDT efficacy of nanophotosensitizers was also higher than Ce6 alone, as shown in Figure 8b, i.e., nanophotosensitizers decreased cell viability with higher efficacy than Ce6 alone. Additionally, Ce6tetraHA and Ce6decaHA nanophotosensitizers have similar efficacy in ROS generation and PDT. The results indicated that Ce6tetraHA or Ce6decaHA nanophotosensitizers have higher ROS production efficacy and PDT efficacy as compared to Ce6 alone.

Pretreatment with free HA was to block the CD44 receptor of cancer cells and this was used to evaluate CD44 receptor targetability of Ce6tetraHA nanophotosensitizer. Red fluorescence color in cancer cells was strongly expressed at cancer cells by the treatment of Ce6tetraHA nanophotosensitizer (Figure 9a). However, pretreatment with free HA against cancer cells significantly decreased the red fluorescence color, which indicated that the Ce6tetraHA nanophotosensitizer can be delivered to the cells through CD44 receptor and blocking of CD44 receptor inhibited the intracellular delivery of Ce6tetraHA nanophotosensitizer. Figure 9b showed the effect of HA pretreatment on the ROS generation and PDT efficacy. Pretreatment with HA for blocking of the CD44 receptor resulted in a decrease of ROS generation and PDT efficacy when compared to CD44 targeting, while Ce6 alone did not respond to CD44 blocking, which indicated that Ce6tetraHA nanophotosensitizers specifically targeted CD44-receptor of cancer cells and PDT efficacy also affected by CD44 receptor.

### 3.4. In Vivo Animal Imaging Study Using Mouse Tumor Xenograft Model

*In vivo* biodistribution of Ce6tetraHA nanophotosensitizer was studied with HCT116 cells. The tumor xenograft model using HCT116 cells was prepared by the subcutaneous injection of HCT116 cells to the back of mice (Figure 10). As shown in Figure 10a, tumor tissues expressed strong fluorescence intensity and, especially, fluorescence intensity was strongest among all organs. These results indicated that nanophotosensitizers efficiently targeted solid tumor. The fluorescence intensity in liver and lung was reduced. As similar to in vitro study, fluorescence intensity in the tumor region was significantly decreased when HA (20 mg/kg) was administered 1h before the injection of nanophotosensitizers to block CD44 receptor of tumor cells (Figure 10b). That is, HA pretreatment efficiently blocked the delivery of nanophotosensitizers against to tumors. In organ distribution, fluorescence intensity was significantly decreased in tumor tissue, while liver expressed the strongest fluorescence when compared to other organs. Practically, the fluorescence density was negligible in tumor tissues as compared to that of liver, which indicates that nanophotosensitizers efficiently targeted the CD44 receptor of tumor cells and their delivery capacity can be controlled by CD44 recognition.

## 4. Discussion

PDT for cancer treatment is believed to be a promising candidate for the improvement of patient’s quality of life due to its safety and low side effect [34,35]. Various photosensitizers have been developed to improve PDT efficacy against cancers [1,2,3,4,5,8,9,34,35,36,37]. Despite the successful application of first or second-generation photosensitizers, photosensitizer itself has drawbacks, such as deficiency of targetability against cancer, limitation in penetration depth in human tissues, low ROS productivity, and low PDT efficacy [3,9,10,11,36,37,38]. Even though Ce6 has higher penetration efficiency than a traditional photosensitizer, such as porphyrin, its delivery capacity and penetration depth in the tumor tissues should still be improved for clinical application. Many scientists investigated novel photosensitizers to improve therapeutic efficacy [8,34,35,36,37,38,39]. In preliminary clinical trials, talaporfin sodium improved the life quality of patient of bile duct carcinoma and maintained patency for 16 month [35]. Furthermore, Kataoka et al., reported that glucose or mannose-conjugated Ce6 has stronger antitumor activity than second-generation talaporfin PDT and they have superiority in tumor targeting when compared to conventional photosensitizers [8]. Furthermore, photosensitizers are regarded as a promising candidate for diagnosis and image-guided surgery of tumors [19,20]. Photosensitizers facilitate the fluorescence detection of tumor by distinguishing them from normal tissues [19]. Subsequently, tumors can be removed by PDT or image-guided surgery while using endoscope [19,20,21]. Nanophotosensitizers as third generation photosensitizers have been investigated in last two decades [8,18,21,38,39]. We previously showed that chitosan nanoparticles significantly increased the penetration efficiency in tumor tissues, longer retention time in tumor, absorption in gastrointestinal region, and PDT efficacy [18]. Since nanomedicines, such as nanoparticles, polymeric micelles, and nanomaterials, have unique properties, such as large surface area, long blood circulation with avoidance of reticuloendothelial (RES) organ uptake, passive/active targeting efficiency against cancer cells, and solubilization of hydrophobic drugs, they possess advantages for the targeting of tumor microenvironment [18,40,41,42,43,44,45]. Among various nanomaterials, hyperbranched polymer materials, such as dendrimer, are spotlighted as a drug-delivery carrier because they possess unique properties, such as lower viscosity, intrinsic molecular behavior, and a lower hydrated radius [45,46,47]. Furthermore, dendrimers have many functional groups for modification with anticancer drugs and targeting moiety [45]. We previously reported that dendrimer nanoparticles significantly improve the intracellular delivery of doxorubicin, overcomes multi-drug resistance and targets solid tumor specifically [45]. Kadhim et al., reported that porphyrin-cored hyperbranched polymers displayed good PDT-induced toxicity against bladder carcinoma cells with no dark toxicity, while porphyrin itself showed significant dark toxicity [47]. Furthermore, Yang et al., also reported that Ce6-incorporated polyethyleneimine nanoparticles have superior PDT efficacy and intracellular delivery capacity [21].

In this study, hyperbranched Ce6 was synthesized while using disulfide linkages and HA also conjugated for tumor targeting, as shown in Figure 1. Hyperbranched Ce6 can be disintegrated by reducing agents, such as GSH or DTT. There is no report for the branched Ce6 material anywhere as far as we know. Most of the nano-scale photosensitizers are related to simple branched polymers, linear polymers, and nanomaterials [15,18,21,39,47]. The higher Ce6 uptake ratio in the cancer cells was observed with Ce6tetraHA or Ce6decaHA nanophotosensitizers with negligible dark cytotoxicity (Figure 6). Furthermore, nanophotosensitizers efficiently targeted cancer cells by redox-sensitivity and HAse-sensitivity. Subsequently, Ce6 release from nanophotosensitizers is controlled by DTT (redox sensitive) and HAse in tumor cells. The increase of GSH concentration and the presence of HAse increased fluorescence intensity of nanophotosensitizer solution (Figure 5). Due to the abnormal microenvironment, tumor tissues have quite different status in physiology and biochemistry, i.e., tumor cells express various carcinogenic proteins and acidic environment of tumor microenvironment contributes to the secretion of various molecules [21,48,49]. For example, the redox-potential in cancer cells is higher than normal cells or tissues and GSH is elevated during oxidative stress [50,51]. Especially, cancer cells have a significantly higher intracellular GSH level than normal cells and these have been considered as a cancer target [29,30,50,51,52]. Since GSH in cancer cells degrades the disulfide bond of conjugates, these unique properties have been applied in tumor targeting [30]. Lee and Jeong approved that the fluorescence intensity of Ce6 was increased by the degradation of disulfide bond in the nanoparticles and then the release of anticancer drug was accelerated [30]. Furthermore, the CD44 receptor is normally over-expressed in the aggressive cancer cells, such as U87MG cells or HCT116 cells, and closely associated with motility on the HA-coated surface [26,53]. These properties can also be applicable in tumor targeting while using HA-bioactive agent conjugates [26,29,54]. The surface of Ce6tetraHA or Ce6decaHA nanophotosensitizers are composed of HA, due to the hydrophilicity of HA and liphophilicity of hyperbranched Ce6. HA on the nanophotosensitizer surfaces interacted with CD44 receptor of U87MG cells (Figure 9a). The blocking of CD44 receptor of U87MG cells efficiently inhibited the internalization of nanophotosensitizer into cancer cells while nanophotosensitizers easily targeted cancer cells without the blocking of CD44 receptor. As shown in Figure 10, the delivery of Ce6tetraHA nanophotosensitizers against tumor tissues was also achieved by interaction with the CD44 receptor of tumor cells, i.e., fluorescence intensity was the strongest in tumor tissues when Ce6tetraHA nanophotosensitizers were i.v. administered. However, CD44 receptor blocking induced decrease in fluorescence intensity, which indicated that the CD44-mediated delivery of nanophotosensitizers was also blocked. Park et al., reported that HA-poly(DL-lactide-co-glycolide) (PLGA) diblock copolymers having disulfide bond delivered anticancer drug through CD44 receptor-mediated pathway and GSH-sensitive manner with minimized side-effects against normal cells [29]. Our results also showed that Ce6tetraHA nanophotosensitizers deliver the Ce6 by CD44 receptor- and GSH- dual sensitive manner.

In conclusion, we synthesized hyperbranched Ce6 conjugates to produce redox-sensitive nanophotosensitizers. Ce6tetraHA nanophotosensitizers have a small diameter of less than 200 nm. The Ce6 release rate was faster in the presence of DTT and HAse, which indicated that Ce6tetraHA/Ce6decaHA nanophotosensitizers have redox-sensitive and HAse-sensitive release properties. Ce6tetraHA/Ce6decaHA nanophotosensitizers showed superior PDT efficacy, i.e., they have higher Ce6 accumulation efficacy, higher ROS productivity, and higher PDT efficacy than Ce6 alone. Furthermore, the delivery of hyperbranched Ce6 nanophotosensitizers against cancer cells was achieved by the recognition of CD44 receptor in vitro and *in vivo*, i.e., CD44 receptor blocking by HA pretreatment efficiently inhibited the tumor targeting of nanophotosensitizers. Fluorescence intensity in the tumor tissues was significantly stronger than that of other organs, while HA pretreatment resulted in the strongest fluorescence intensity in liver with negligible intensity in the tumor tissues. These results indicated that the delivery of nanophotosensitizers against cancer cells was controlled by redox-sensitive and CD44 receptor-responsive manner.

## Figures and Tables

**Figure 1 materials-12-03080-f001:**
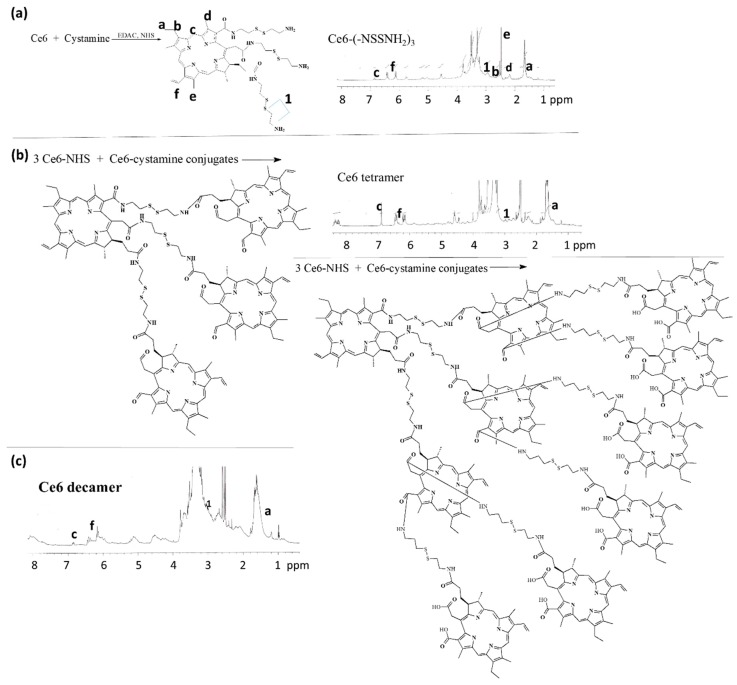
Synthesis and ^1^H NMR spectra of Ce6-cystamine (**a**), Ce6 tetramer (**b**), and Ce6 decamer (**c**). For measurement of ^1^H NMR (500 mHz) spectra, Ce6, Ce6 tetramer, and Ce6 decamer were dissolved in deuterated dimethylsulfoxide (DMSO) (DMSO d_6_).

**Figure 2 materials-12-03080-f002:**
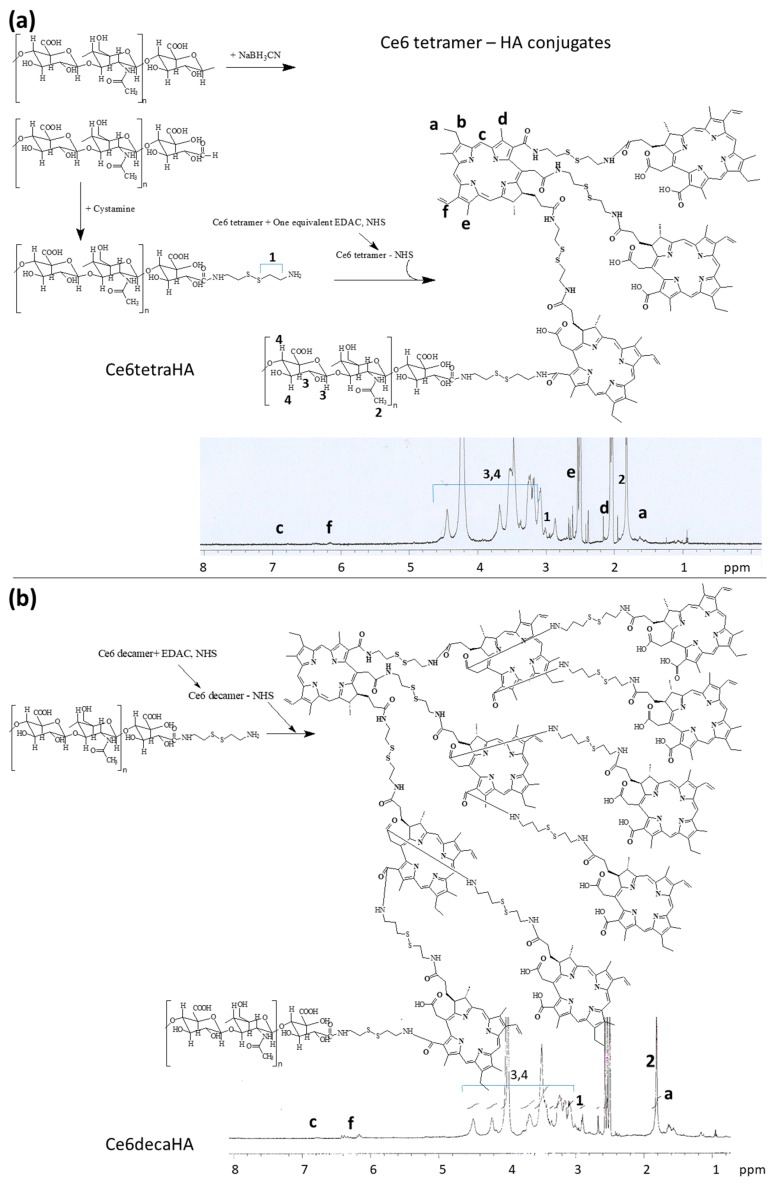
Synthesis and ^1^H NMR spectra of (**a**) Ce6tetraHA conjugates and (**b**) Ce6decaHA conjugates. For measurement of ^1^H NMR (500 mHz) spectra, Ce6tetraHA and Ce6decaHA conjugates were distributed in 0.2 mL of D_2_O and then mixed with 0.8 mL DMSO-d_6_.

**Figure 3 materials-12-03080-f003:**
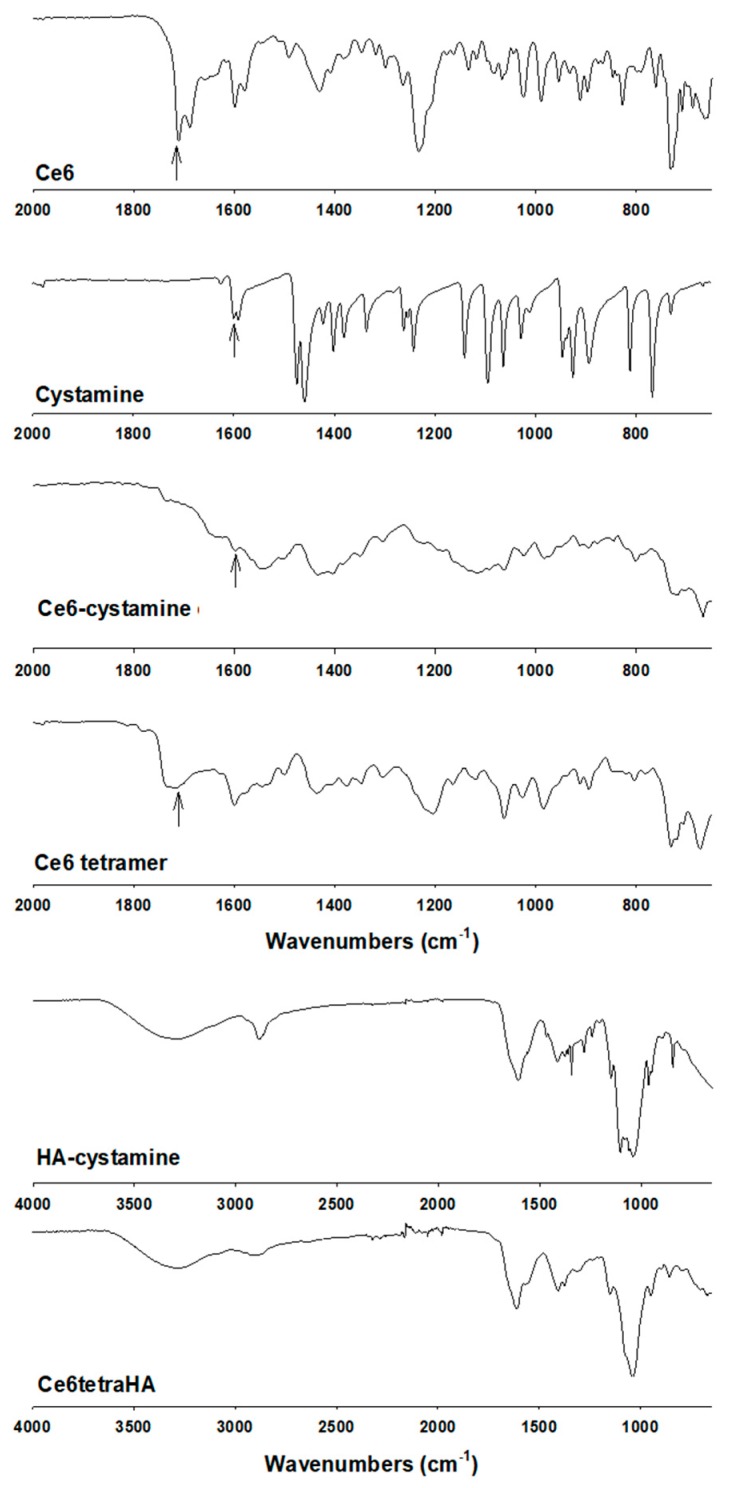
FT-IR spectra of Ce6tetraHA.

**Figure 4 materials-12-03080-f004:**
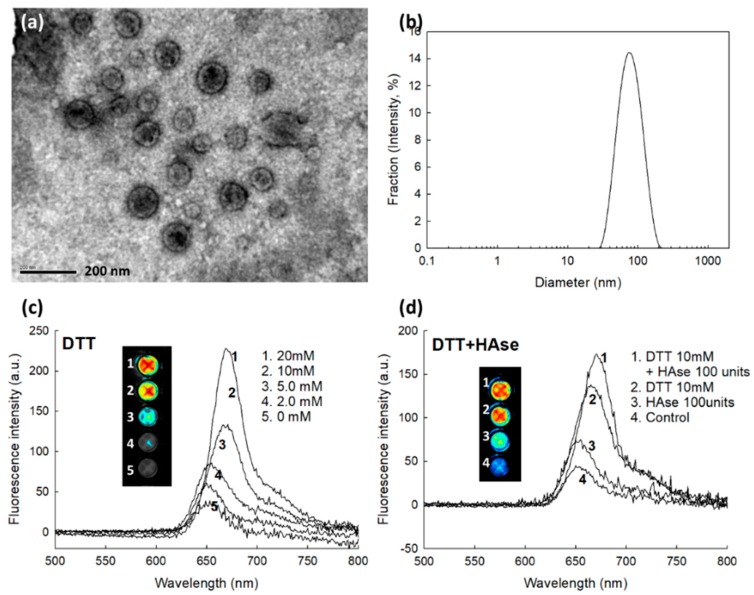
Morphological observation (**a**) and particle size distribution (**b**) of Ce6tetraHA nanophotosensitizers (Average particle size was 82.6 ± 5.6 nm). The effect of 1,4-dithiothreitol (DTT) (**c**) and DTT/hyaluronidase (HAse) (**d**) on the fluorescence spectra of Ce6tetraHA nanophotosensitizers. For DTT treratment, various concentrations of DTT in PBS was incubated with Ce6tetraHA nanophotosensitizers at 37 °C for 3 h and then measured fluorescence spectra. For HAse treatment, 100 units of HAse in PBS were incubated with nanophotosensitizer for 3 h at 37 °C. Final concentration of Ce6tetraHA nanophotosensitizers was adjusted as 0.5 mg/mL.

**Figure 5 materials-12-03080-f005:**
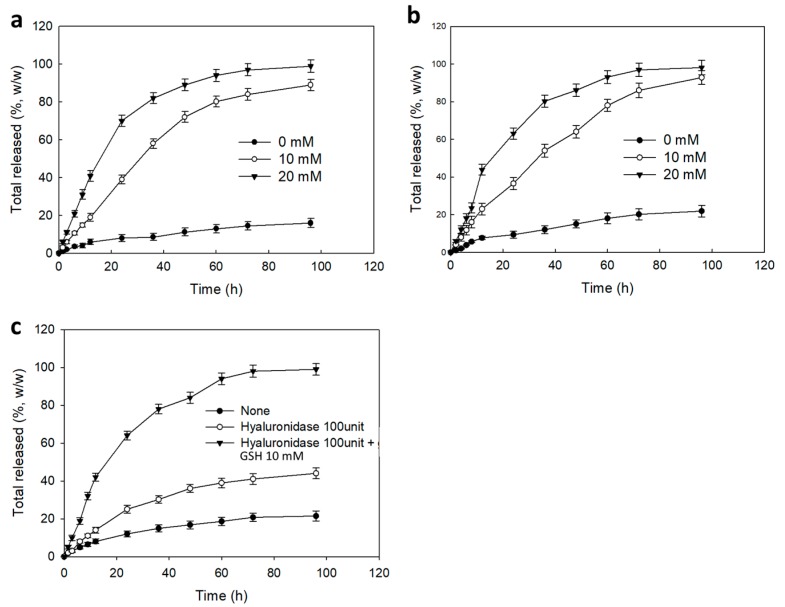
The effect of DTT on the Ce6 release dynamics from Ce6tetraHA (**a**) and Ce6decaHA (**b**) nanophotosensitizers. (**c**) The effect of HAse/DTT on the Ce6 release dynamics from Ce6tetraHA nanophotosensitizers. All the values are average ± S.D. from three different experiments.

**Figure 6 materials-12-03080-f006:**
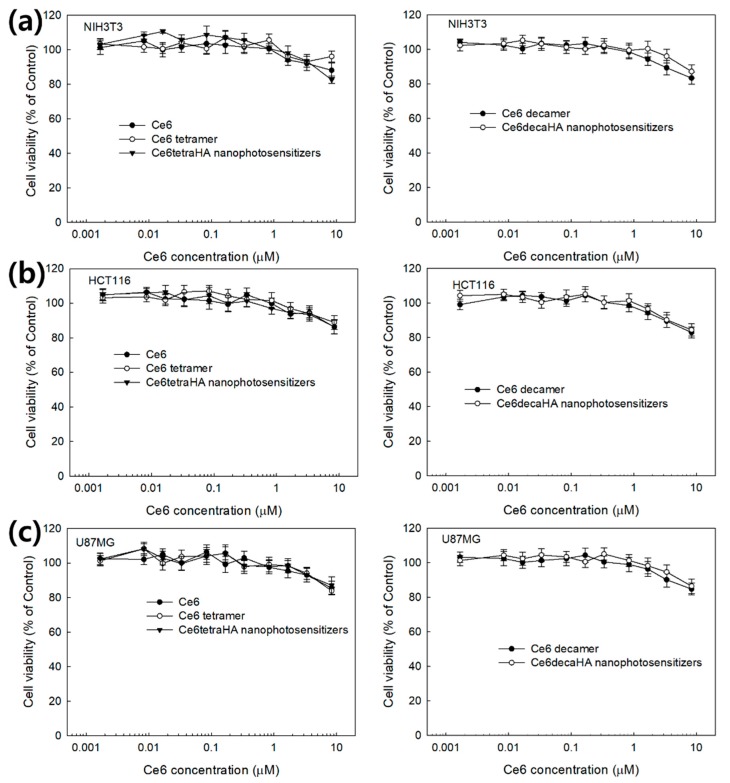
Dark toxicity of Ce6, Ce6tetraHA and Ce6decaHA nanophotosensitizers against NIH3T3 mouse fibroblast cells (**a**), HCT116 human colon carcinoma cells (**b**) and U87MG human malignant glioma cells (**c**). All the values are average ± S.D. from results of a single independent experiment with eight replicate.

**Figure 7 materials-12-03080-f007:**
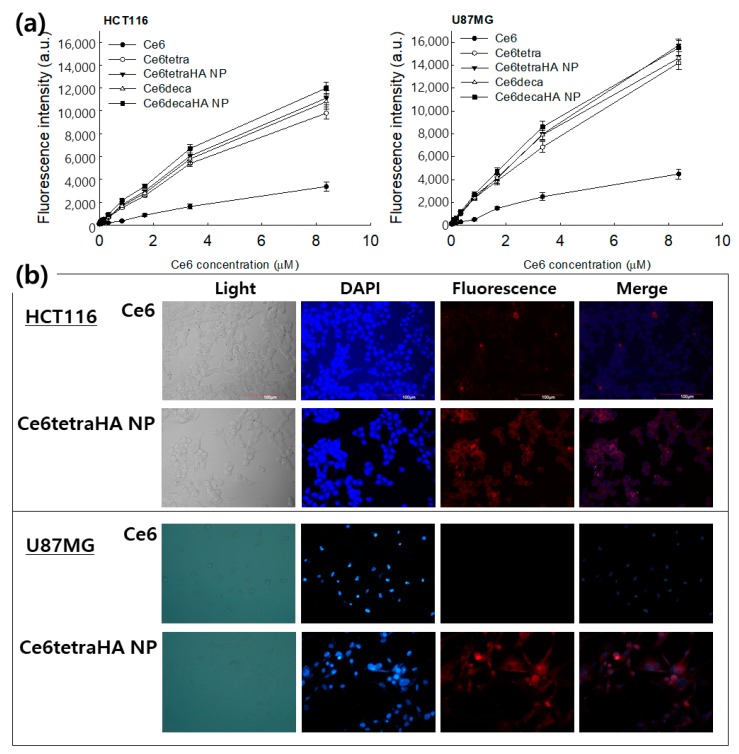
Ce6 uptake ratio (**a**) and fluorescence images (**b**) in the HCT116 cells and U87MG cells. For fluorescence images, Ce6 concentration was adjusted in 3.35 μM (2.0 μg/mL) and exposed to cells for 90 min. All the values are average ± S.D. from results of a single independent experiment with eight replicate.

**Figure 8 materials-12-03080-f008:**
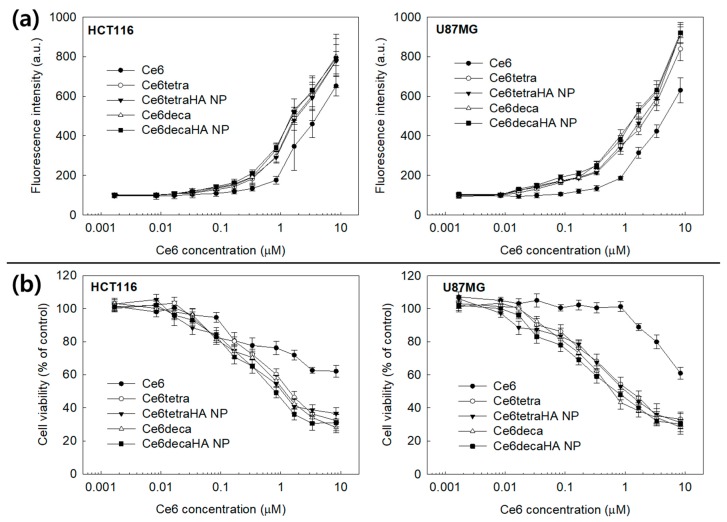
The effect of Ce6 or Ce6tetraHA and Ce6decaHA nanophotosensitizer treatment against cancer cells. Reactive oxygen species (ROS) generation (**a**) and hotodynamic therapy (PDT) treatment (**b**). Cells were irradiated at 664 nm (2 J/cm^2^). All values are average ± S.D. from results of a single independent experiment with eight replicate.

**Figure 9 materials-12-03080-f009:**
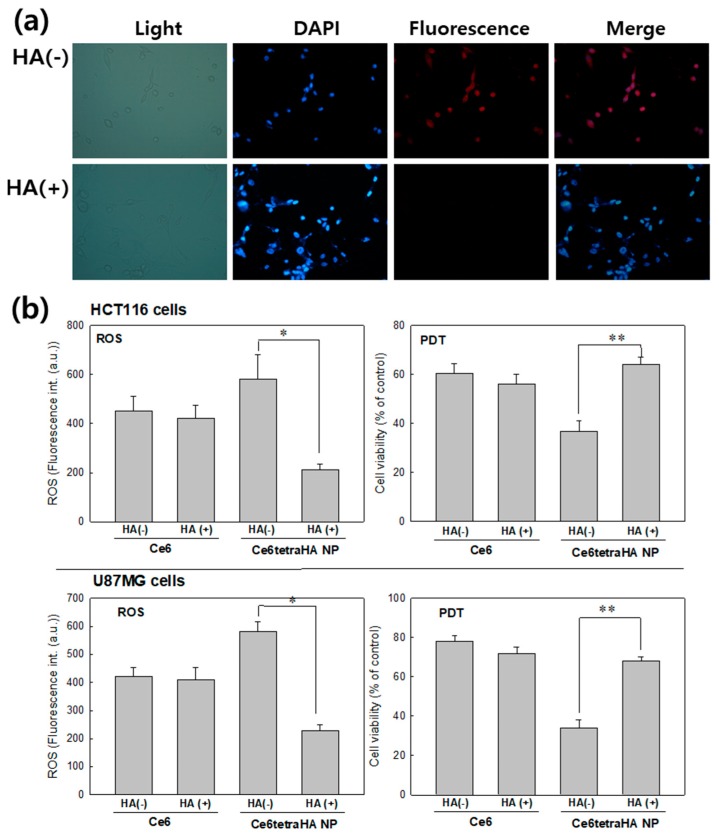
(**a**) The effect of blocking of CD44 receptor. Fluorescence images of U87MG cells. For blocking of CD44 receptor, 10 times higher amount of free HA was pre-treated for 60 min. before treatment of Ce6 or nanophotosensitizers. (**b**) ROS generation and PDT effect on the cancer cells. For fluorescence images, Ce6 concentration was adjusted to 3.35 μM (2.0 μg/mL) and exposed to cells for 60 min. All values are average ± S.D. from results of a single independent experiment with eight replicate. *; *p* < 0.001. **; *p* < 0.001.

**Figure 10 materials-12-03080-f010:**
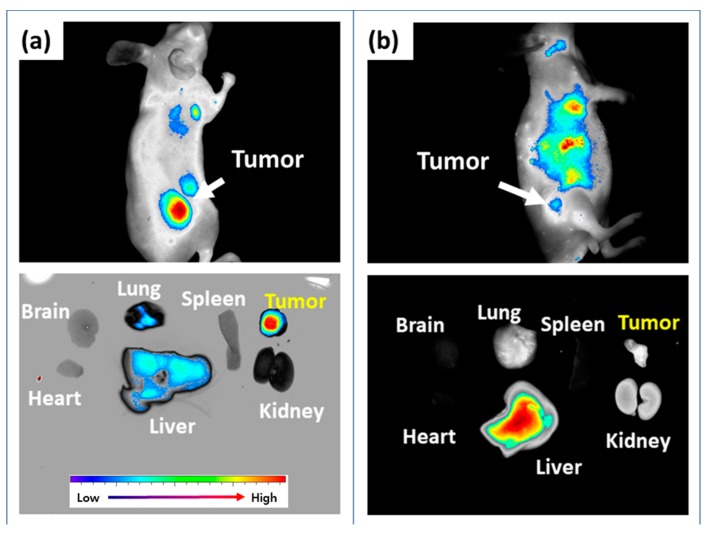
*In vivo* animal imaging using HCT116-bearing tumor xenograft mouse model. (**a**) Fluorescence imaging of tumor xenograft of HCT116 tumor. (**b**) The effect of pretreatment of HA on the fluorescence imaging of mouse. Ce6tetraHA nanophotosensitizers were i.v. administered (5.0 mg/kg) and then mice were observed under fluorescence 1 day later. For pretreatment of HA, HA (20 mg/kg) were i.v. administered 1h before nanophotosensitizer administration.

**Table 1 materials-12-03080-t001:** Particle size of nanophotosensitizers.

Types of Nanophotosensitizers	Ce6 Contents (%, w/w)	Particle Size (nm) ^a^
Ce6tetraHA NP	22.1	82.6 ± 5.6
Ce6decaHA NP	39.8	181.4 ± 12.3

^a^ Particle size was average ± S.D. from three different measurements. NP = Nanophotosensitizers.

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
