# Peer review of "Hyaluronic Acid-Conjugated with Hyperbranched Chlorin e6 Using Disulfide Linkage and Its Nanophotosensitizer for Enhanced Photodynamic Therapy of Cancer Cells"

_materials, 2019, doi:10.3390/ma12193080_

Round 1

Reviewer 1 Report

In the present work two new nanophotosensitizers were synthesized and tested. Despite the huge number of works studying new photosensitizers this one seems to me very original and interesting. The manuscript is written logically. The methods chosen appropriate. The obtained results will have both applied (new substance with good potential) and fundamental importance (development of nanoparticle creation strategies).In my opinion, the manuscript should be published, but before it is necessary to correct minor flaws.

Line 224-225: It is necessary to correctly indicate the concentration of the antibiotic. 1% is the volume concentration of the solution, the concentration of which is not specified.

Section 2.8: The PDT treatment procedure described very incomplete.  It does not allow to understand: 1) How much solvent (DMSO) was in cell culture during treatment? 2) Was there an equal amount of DMSO among experimental variant with different concentrations of the tested agent? 3) How long have cell cultures been irradiated with light?

Figures 6 and 8: Why are dark toxicity data presented on a logarithmic scale, but phototoxicity data on a linear scale? They are difficult to compare. Moreover, from Figure 8, due to the linear scale, it is not clear at what concentration the photodynamic effect is started. The difference between the minimum active concentrations in the light and dark conditions is one of the most important indicators of the potential effectiveness of a photosensitizer. I am sure that these dependencies of all andpoints on concentration in this case should be shown on a logarithmic scale.

The particles size are specified in the text. So, I do not see the sense in table 1, which do not show any additional information.

To make it easier to understand the difference between the data presented in Fig. 4 and Fig. 5 you can add the word "Dynamics" to the title of the Fig.5. Similar clarification should be added in lines 362-369.

Comparison of the biological activities of new compounds with Ce6 through mass concentration in this case is really optimal, however, in order to make it easier to compare the biological testing data with similar ones from other works, it seems to me better to indicate the molar concentration of Ce6 after the mass one in brackets one or several times in manuscript.

Figure 10: Is it possible to show the ratio of concentrations (fluorescence) in the tissues? The "high/low" gradation does not provide an information about the relative range of observed differences.

Section 2.10: "... were exposed to Ce6 or nanophotosenstitizer for 90 min" but Figure 7: "... exposed to cells for 60 min".

Line 46: "resultd"

Line 491 "or or"

Line 492 "decorated" - not a very good word

Author Response

Answer to Reviewer 1’s comment

In the present work two new nanophotosensitizers were synthesized and tested. Despite the huge number of works studying new photosensitizers this one seems to me very original and interesting. The manuscript is written logically. The methods chosen appropriate. The obtained results will have both applied (new substance with good potential) and fundamental importance (development of nanoparticle creation strategies).In my opinion, the manuscript should be published, but before it is necessary to correct minor flaws.

* Line 224-225: It is necessary to correctly indicate the concentration of the antibiotic. 1% is the volume concentration of the solution, the concentration of which is not specified.

Answer) Thanks for your comment. According to your comment, we corrected this as follows:

supplemented with 10% (v/v) heat-inactivated fetal bovine serum (FBS) (Invitrogen) and 1% (v/v) antibiotics at 37°C in a 5% CO2 incubator. HCT116 cells were cultured in RPMI1640 (Gibco, Grand Island, NY, USA) supplemented with 10% (v/v) heat-inactivated fetal bovine serum (FBS) (Invitrogen) and 1% (v/v) antibiotics.

* Section 2.8: The PDT treatment procedure described very incomplete.  It does not allow to understand: 1) How much solvent (DMSO) was in cell culture during treatment? 2) Was there an equal amount of DMSO among experimental variant with different concentrations of the tested agent? 3) How long have cell cultures been irradiated with light?

Answer) Thanks for your comment.

(1) The concentration of DMSO was adjusted below 0.5 % (v/v) during cell culture experiment.

 (2) Practically, less than 0.5 % (v/v) DMSO was not affected to the viability or growth of cells. For Ce6 treatment, DMSO concentration was adjusted to 0.5 % (v/v) and diluted with media. However, DMSO was not used for nanophotosensitizers because nanophotosensitizers will not maintain nanoparticle morphology in DMSO. Instead of DMSO, 1.2 mm syringe filters were used for nanophotosnesitizers. Anyway, we revised the manuscript in this issues as follows;

For Ce6 treatment, Ce6 was dissolved in DMSO and then diluted at least one hundred times with media. Final concentration of DMSO in cell culture was adjusted below 0.5 % (v/v). For treatment of hyperbranched Ce6 treatment, Ce6 tetramer or Ce6 decamer was dissolved in DMSO and the diluted (Final DMSO concentration: 0.5 % (v/v)).

100 µl serum-free fresh media was added to cells and cells were exposed to 664 nm light using expanded homogenous beam (SH Systems, Gwangju, Korea). The dose of light irradiation was 2.0 J/cm2. The light intensity was measured with a photo radiometer (Delta Ohm, Padua, Italy) and cells were irradiated for 9 min 29 sec for 2.0 J/cm2 in our system.

* Figures 6 and 8: Why are dark toxicity data presented on a logarithmic scale, but phototoxicity data on a linear scale? They are difficult to compare. Moreover, from Figure 8, due to the linear scale, it is not clear at what concentration the photodynamic effect is started. The difference between the minimum active concentrations in the light and dark conditions is one of the most important indicators of the potential effectiveness of a photosensitizer. I am sure that these dependencies of all andpoints on concentration in this case should be shown on a logarithmic scale.

Answer) Thanks for your comment. According to your comment, we corrected the scale of Figure 6 and 8.

* The particles size are specified in the text. So, I do not see the sense in table 1, which do not show any additional information.

Answer) Thanks for your comment. According to your comment, we specified in the text in Results section and give additional information in Figure 4 legend and Table 1.

In Exprimental section

To estimate average particle size, they were measured particle sizes at least three times and expressed as average±S.D..

In Results section

Figure 4(b) showed that the particle size of Ce6tetraHA nanophotosensitizers revealed mono-dispersed pattern with small particle size distribution and particle sizes were less than 200 nm. As shown in Table 1, average particle sizes of Ce6tetraHA or Ce6decaHA nanophotosensitizers were 82.6±5.6 nm and 181.4±12.3 nm from three different measurements, respectively.

In Table 1

Table 1. particle size of nanophotosensitizers

Ce6 contents (%, w/w)

Particle size (nm) a

Ce6tetraHA NP

Ce6decaHA NP

22.1

39.8

82.6±5.6

181.4±12.3

a Particle size was average±S.D. from three different measurements.

NP = nanophotosensitizers

In figure 4

Figure 4. Morphological observation (a) and particle size distribution (b) of Ce6tetraHA nanophotosensitizers (Average particle size was 82.6±5.6 nm). The effect of DTT (c) and DTT/HAse (d) on the fluorescence spectra of Ce6tetraHA nanophotosensitizers. For DTT treratment, various concentrations of DTT in PBS was incubated with Ce6tetraHA nanophotosensitizers at 37oC for 3h and then measured fluorescence spectra. For HAse treatment, 100 units of HAse in PBS was incubated with nanophotosensitizer for 3h at 37oC. Final concentration of Ce6tetraHA nanophotosensitizers was adjusted as 0.5 mg/ml.

* To make it easier to understand the difference between the data presented in Fig. 4 and Fig. 5 you can add the word "Dynamics" to the title of the Fig.5. Similar clarification should be added in lines 362-369.

Answer) Thanks for your comment. According to your comment, we revised the manuscript.

In Figure 5

Figure 5. The effect of DTT on the Ce6 release dynamics from Ce6tetraHA (a) and Ce6decaHA (b) nanophotosensitizers. (c) The effect of HAse/DTT on the Ce6 release dynamics from Ce6tetraHA nanophotosensitizers.

In results section

Figure 5 showed the Ce6 release dynamics from the nanophotosensitizers. As shown in Figure 5(a) and (b), Ce6 release dynamics from nanophotosensitizers was slow in the absence of DTT. However, Ce6 release dynamics became significantly higher in the presence of DTT and release dynamics was gradually increased according to the increase of DTT concentration. Furthermore, the addition of HAse also increased the release dynamics of Ce6 as shown in Figure 5(c). Especially, dual addition of HAse and DTT also significantly increased the release dynamics of Ce6 from nanophotosensitizers. These results indicated that Ce6 release dynamics from nanophotosensitizers can be controlled by redox-sensitive manner and by HAse.

* Comparison of the biological activities of new compounds with Ce6 through mass concentration in this case is really optimal, however, in order to make it easier to compare the biological testing data with similar ones from other works, it seems to me better to indicate the molar concentration of Ce6 after the mass one in brackets one or several times in manuscript.

Answer) Thanks for your comment. According to your comment, we revised the manuscript. We changed the concentration unit from microgram/ml to microM In Figure 6, 7, and 8 as follows:

In Figure 6

In Figure 7

In Figure 8

In Results

Both Ce6 and nanophotosensitizers have little toxicity until 8.375 mM (5 mg/ml) against NIH3T3 cells and cancer cells in the absence of light, i.e. viability of cells was maintained higher than 80 % until 8.375 mM (5 mg/ml) at all formulations.

In Figure 7 legend

Figure 7. Ce6 uptake ratio (a) and fluorescence images (b) in the HCT116 cells and U87MG cells. For fluorescence images, Ce6 concentration was adjusted in 3.35 mM (2.0 mg/ml) 2 mg/ml and exposed to cells for 90 min.

In Figure 9

Figure 9. The effect of blocking of CD44 receptor. (a) Fluorescence images of U87MG cells. (b) ROS generation and PDT effect on the cancer cells. For fluorescence images, Ce6 concentration was adjusted to 3.35 mM (2.0 mg/ml) and exposed to cells for 60 min. For blocking of CD44 receptor, 10 times higher amount of free HA was pre-treated for 60 min before treatment of Ce6 or nanophotosensitizers. *,**; p < 0.001.

* Figure 10: Is it possible to show the ratio of concentrations (fluorescence) in the tissues? The "high/low" gradation does not provide an information about the relative range of observed differences.

Answer) Thanks for your comment. Actually, equipment for fluorescence images do not provide an absolute value of fluorescence images. Then we couldn’t obtain density of fluorescence images of each organ. We can only shows fluorescence images in the Figure 10.

* Section 2.10: "... were exposed to Ce6 or nanophotosenstitizer for 90 min" but Figure 7: "... exposed to cells for 60 min".

Answer) Thanks for your comment. According to your comment, we revised the manuscript. That is a miss-typing. All experiment in Figure 7, cells were exposed to Ce6 or nanophotosensitizers for 90 min.

In Figure 7 legend

Figure 7. Ce6 uptake ratio (a) and fluorescence images (b) in the HCT116 cells and U87MG cells. For fluorescence images, Ce6 concentration was adjusted in 2 mg/ml and exposed to cells for 90 min.

* Line 46: "resultd"

Line 491 "or or"

Line 492 "decorated" - not a very good word

Answer) Thanks for comments. According to your comment, we corrected the mistyping.

Line 46: à "resulted"

Line 491 à " or"

Line 492 "decorated" - not a very good word à Ce6tetraHA or Ce6decaHA nanophotosensitizers are composed of HA on the surface of nanoparticles.

Reviewer 2 Report

This manuscript describes the synthesis of two novel nanomaterials and their characterization as potential photosensitizers for PDT. Below is a list of issues raised by the manuscript, as they appeared in the text.

 This manuscript describes the synthesis of two novel nanomaterials and their characterization as potential photosensitizers for PDT. Below is a list of issues raised by the manuscript, as they appeared in the text.

1- Title: In what sense are the new materials biodegradable? And how was this “biodegradability” assessed?

2- lns 39-40: What is the meaning of “redox-sensitive and CD44-sensitive photodynamic therapy”? Figure 5 shows that DTT, GSH and hyaluronidase significantly increase the release of Ce6 from the nanomaterials, but is this release advantageous? The results presented in this manuscript do not suggest so.

3- ln 49: “resulted in higher intracellular Ce6 accumulation”. How did the authors discriminate between Ce6 accumulation and Ce6tetraHA or Ce6decaHA accumulation? Also, the two nanoparticles under consideration are abbreviated in several different ways throughout the manuscript (including figures). Please, ensure consistency.

4- lns 50-51: “Ce6tetraHA 50 nanophotosensitizers were delivered by CD44 receptor-mediated manner”. Do the authors have any suggestion regarding the molecular mechanisms of this “CD44 receptor-mediated manner”?

5- ln 55: “We suggest Ce6tetraHA as a promising candidate”. Why not Ce6decaHA also?

6- ln 67: What is the meaning of “deeper penetration efficiency”? + “low penetration depth in the tumor”. Please, be aware that the “penetration depth” (ln 68) refers to the light employed, not to the photosensitizer.

7- lns 71-85: The authors mention several Ce6 derivatives that were synthesized and evaluated as photosensitizers for PDT, but then, in the Discussion section, they do not compare their PDT potential with those of the nanoparticles under evaluation in their study. Also, is there any reason why they have not yet been introduced in the clinic?

8- In the experimental section, why is the characterization of the nanoparticles (section 2.3) presented BEFORE their “fabrication” (section 2.4)?

9- Section 2.7: Please, specify the source of the NIH3T3 cells used and describe how they were grown.

10- Legend to Figure 5: (c) does NOT describe “the effect of HAse/DTT”. Also, the error bars in this and other figures suggests multiple experiments. If this is the case, please specify the number of experiments performed.

11- Figure 5 shows a significant release of Ce6 from the nanoparticles, even in the absence of DTT, GSH or HAse? Can this instability compromise the use of these particles in the clinic? I.e., do they need to be prepared immediately before administration? How feasible is this?

12- Why were normal cells NOT employed in the PDT studies? It would have been important, as one of the main limitations of PDT is photosensitization. Please, provide data. The same applies to the uptake experiments.

13- In the in vitro studies, Ce6 was used for comparison purposes. Why was it not done in the in vivo studies?

Other issues:

- The manuscript contains several awkward sentences that should be rephrased, to improve the manuscript’s readability. Below are some examples:

* lns 64-65: “and then non-toxic reaction is expected in the normal tissues or cells [5-7]”.

* lns 72-73: “Ce6-polyvinylpyrrolidone (PVP) formulations revealed less in vivo intrinsic phototoxicity”. This sentence is confusing. A lower phototoxicity should not be viewed as an advantage in the context of PDT.

* lns 75-76: “nanophotosensitizers composed of poly(ethylene glycol) (PEG)-Ce6 conjugates showed activated state in aqueous solution”. What is the meaning of “showed activated state”?

* lns 81-82: “Nanomedicine-based photodynamic therapy also enables disease diagnosis and image-guided surgery [19-21].” How can PDT be used in diagnosis and surgery?

* ln 89: What is the meaning of higher redox state?

* lns 87-89: “Compared to normal cells and tissues, tumor cells express various receptors with higher degree, higher enzymatic activity such as hyaluronidase (HAse), unbalanced immune system, poor perfusion, acidic pH environment and higher redox-status [22-24].” It is important to distinguish the tumor cells from the tumor microenvironment. Tumor cells do NOT express and unbalanced immune system.

* lns 97-98: What is the meaning of “high metabolic activity in tumor tissues promote reduction/oxidation activities”?

* ln 107-108: The sentence “HA has hydrophilic macromolecules” is confusing.

* ln 244: “Ce6 or nanophotosensitizers were exposed to cells”. It is the cells that are exposed to the nanophotosensitizers, not the other way round.

* lns 263-264: “When the solid tumor were reached larger than 4 mm in diameter”. This sentence is confusing.

* lns 386-389: The sentences “Figure 7 showed the comparison of Ce6 and nanophotosensitizers in the Ce6 uptake ratio in HCT116 cells and U87MG cells in vitro.” and “all chemicals showed that Ce6 uptake ratio at HCT116 cells and U87MG cells was gradually increased according to the increase of Ce6 concentration.” are very confusing.

* lns 401-402: Please, improve the sentence “differences in ROS generation and PDT efficacy was practically small between Ce6tetraHA and Ce6decaHA nanophotosensitizers”.

* ln408: “HA was pretreated to cells”. Please, improve.

* lns 412-413: “Ce6tetraHA nanophotosensitizer can be delivered to the cells through CD44 receptor”. This sentence suggests that molecules may enter cells through receptors, whereas they do so through transporters.

* lns 479-480: “Ce6tetraHA nanophotosensitizers displayed redox-sensitive and HAse-sensitive behavior in photosensitizer release”. This sentence is not very clear.

* lns 485-486: “this peculiarity have been considered as a cancer target of bioactive molecules [28,50,51]”: Confusing.

* lns 488-490: “we approved that Ce6 can be activated through degradation of disulfide bond by GSH treatment to nanoparticles and then accelerated drug release with improved cancer cell toxicity [28].” Very confusing. For instance, what is the meaning of “activated” in this context?

- It is advisable to use the use the 'spell-check' and 'grammar-check' functions of the word processor, to minimize spelling and grammar (namely in terms of verb conjugation) errors.

Author Response

Answer to Reviewer 2’s comment

This manuscript describes the synthesis of two novel nanomaterials and their characterization as potential photosensitizers for PDT. Below is a list of issues raised by the manuscript, as they appeared in the text.

1- Title: In what sense are the new materials biodegradable? And how was this “biodegradability” assessed?

Answer) Thanks for your comments. We used “biodegradable” in the title because HA can be degradable by hyaluronidase and Ce6 tetramer/Ce6decamer can be divided to Ce6 monomer by GSH or DTT. Then all components of Ce6tetraHA NP or Ce6decaHA NP can be degraded to monomeric units. Then we used “biodegradable.

2- lns 39-40: What is the meaning of “redox-sensitive and CD44-sensitive photodynamic therapy”? Figure 5 shows that DTT, GSH and hyaluronidase significantly increase the release of Ce6 from the nanomaterials, but is this release advantageous? The results presented in this manuscript do not suggest so.

Answer) Thanks for your comments. In fact, we used redox-sensitive due to that disulfide linkage can be degraded by DTT or GSH (GSH is an anti-oxidant material, which is abundant in cancer cells.). And this linkage can be disintegrated by GSH in cancer cells. Then we used redox-sensitive because reduction/oxidation potential of cancer cells is significantly higher than normal cells. Furthermore, due to the GSH concentration in cancer cells, Ce6 release rate must be accelerated in the cancer cells because GSH content in the cancer cells are higher than that in normal cells (at least 100 times higher). Anyway, we discussed more in Discussion section and added ref.

For example, redox-potential in cancer cells is higher than normal cells or tissues and GSH is elevated during oxidative stress [50,51]. Especially, intracellular GSH level in cancer cells is significantly higher than normal cells and this peculiarity has been considered as a cancer target of bioactive molecules [27,28,50-52].

Gamcsik, M.P.; Kasibhatla, M.S.; Teeter, S.D.; Colvin, O.M. Glutathione levels in human tumors. Biomarkers. 2012, 17, 671-691.

Bansal, A.; Simon, M.C. Glutathione metabolism in cancer progression and treatment resistance. J. Cell Biol. 2018, 217, 2291-2298.

3- ln 49: “resulted in higher intracellular Ce6 accumulation”. How did the authors discriminate between Ce6 accumulation and Ce6tetraHA or Ce6decaHA accumulation? Also, the two nanoparticles under consideration are abbreviated in several different ways throughout the manuscript (including figures). Please, ensure consistency.

Answer) Thanks for your comments. As shown in Experimental section (2.4. fabrication of Ce6tetraHA/Ce6decaHA nanophotosensitizers), Ce6 contents in Ce6tetraHA NP or Ce6decaHA NP were measured by degradation of nanophotosensitizers with DTT as shown in Experimental section (2.4. fabrication of Ce6tetraHA/Ce6decaHA nanophotosensitizers). Cells were treated with Ce6tetraHA and Ce6decaHA nanophotosensitizers based on Ce6 contents. Anyway, we revised the manuscripts in the Experimental section and Results section (in Table 1).

In Experimental section (2.4. fabrication of Ce6tetraHA/Ce6decaHA nanophotosensitizers).

Ce6 contents in Ce6tetraHA nanophotosensitizers and Ce6decaHA nanophotosensitizers were 22.1 % (w/w) and 39.8 % (w/w), respectively.

Results section

Table 1. particle size of nanophotosensitizers

Ce6 contents (%, w/w)

Particle size (nm) a

Ce6tetraHA NP

Ce6decaHA NP

22.1

39.8

82.6±5.6

181.4±12.3

a Particle size was average±S.D. from three different measurements.

NP = nanophotosensitizers

4- lns 50-51: “Ce6tetraHA 50 nanophotosensitizers were delivered by CD44 receptor-mediated manner”. Do the authors have any suggestion regarding the molecular mechanisms of this “CD44 receptor-mediated manner”?

Answer) Thanks for your comments. Practically, we depicted in Graphical abstract. CD44 receptor-mediated manner means that nanoparticles can be internalized through specific interaction between HA (of nanoparticles) and CD44 receptor of cancer cll surface.   CD44 (known as hyaluronic acid receptor) is a cell-surface glycoprotein involved in cell–cell interactions, cell adhesion and migration. Especially, CD44 is over-expressed in cancer cell surface and associated with motility of cancer cells on the HA-involved ECM materials (Afify A, Purnell P, Nguyen L. Role of CD44s and CD44v6 on human breast cancer cell adhesion, migration, and invasion. Exp Mol Pathol. 2009, 86, 95-100.). Many investigators reported that HA-decorated nanoparticles can be internalized into cancer cells via CD44 receptor of cancer cell surface (Platt VM, Szoka FC Jr. Anticancer therapeutics: targeting macromolecules and nanocarriers to hyaluronan or CD44, a hyaluronan receptor. Mol Pharm. 2008 Jul-Aug;5(4):474-86.). Practically, HA on the nanoparticle surface interacts with CD44 receptor on the surface of cell membrane and then internalized into cells by formation of endosome. Anyway, we revised the manuscript and added more discussion in the Discussion section.

Furthermore, CD44 receptor is normally over-expressed in the aggressive cancer cells such as U87MG cells or HCT116 cells and closely associated with motility on the HA-coated surface [24,53]. These properties also can be applicable in tumor targeting of bioactive agents [24,27,54] by interaction with CD44 receptor of cancer cells. Surface of Ce6tetraHA or Ce6decaHA nanophotosensitizers are composed of HA since HA is a hydrophilic molecule and hyperbranched Ce6 is a liphophilic molecule. HA on the nanophotosensitizers are interacted with CD44 receptor of U87MG cells as shown in Figure 9(a), i.e. internalization of nanophotosensitizers into cancer cells were inhibited by blocking of CD44 receptor while strong red fluorescence was observed without pre-treatment of HA (HA(-)). Ce6tetraHA nanophotosensitizers were also delivered by CD44 receptor-mediated manner in vivo tumor xenograft model as shown in 10, i.e. fluorescence intensity was strongest in tumor tissues when Ce6tetraHA nanophotosensitizers were i.v. administered.

5- ln 55: “We suggest Ce6tetraHA as a promising candidate”. Why not Ce6decaHA also?

Answer) Thanks for your comments. We revised the manuscript according to your comment.

We suggest Ce6tetraHA and Ce6decaHA nanophotosensitizers as a promising candidate for PDT of cancers.

6- ln 67: What is the meaning of “deeper penetration efficiency”? + “low penetration depth in the tumor”. Please, be aware that the “penetration depth” (ln 68) refers to the light employed, not to the photosensitizer.

Answer) Thanks for your kind comment. Practically, Ce6 can be penetrated in the tissue deeper than traditional photosensitizers (such as 5-aminolevulinic acid) but it should be still improved in delivery capacity against tumor tissues. Regarding for “low penetration depth in the tumor” I mean that penetration capacity of photosensitizers in tumor tissues should be still improved even though Ce6 has higher penetration efficacy than traditional photosensitizers. Also, longer wavelength of light can penetrated tissues deeper than short wavelength of light. Anyway, we discussed more in the introduction/Discussion part and revised the manuscript.

In Introduction part

Ce6 has been extensively investigated for PDT of cancer because it has deeper tissue penetration efficiency and is activated at higher wavelength of light than traditional photosensitizers such as 5-aminolevulic acid (5-ALA) or porphyrins, [8,9]. Also, higher wavelength of light can penetrates tissues deeper than lower wavelength of light [6-9]. Despite of these advantages, most of photosensitizers including Ce6 has still low cancer cell specificity and low delivery capacity against the tumor tissues and requirement for long-term sun-shade [3,9-11]. These drawbacks still limit clinical application of photosensitizers [2,12].

In Discussion part

Even though Ce6 has higher penetration efficiency than traditional photosensitizer such as porphyrin, its delivery capacity and penetration depth in the tumor tissues should be still improved for clinical application.

7- lns 71-85: The authors mention several Ce6 derivatives that were synthesized and evaluated as photosensitizers for PDT, but then, in the Discussion section, they do not compare their PDT potential with those of the nanoparticles under evaluation in their study. Also, is there any reason why they have not yet been introduced in the clinic?

Answer) Thanks for your kind comment. Practically, Ce6 is relatively neww-generation of photosensitizer compared to porphyrin or 5-aminolevulic acid derivatives. Then their clinical application is still limited. Fro example, talaporfin sodium (mono-L-aspartyl chlorin e6, NPe6) is typical chlorin e6 derivatives and used in clinics for cancer patients. Furthermore, other kind of chlorin e6 derivatives also investigated by Japanese group. According to your comment, we discussed more for Ce6 derivatives in Discussion section.

Even though Ce6 has higher penetration efficiency than traditional photosensitizer such as porphyrin, its delivery capacity and penetration depth in the tumor tissues should be still improved for clinical application. Many scientists investigated novel photosensitizers to improve therapeutic efficacy [33-38]. In a preliminary clinical trials, talaporfin sodium improved life quality of patient of bile duct carcinoma and maintained patency for 16 month [33]. Furthermore, Kataoka et al., reported that glucose or mannose-conjugated Ce6 has stronger antitumor activity than second-generation talaporfin PDT and they has superiority in tumor targeting compared to conventional photosensitizers [37].

8- In the experimental section, why is the characterization of the nanoparticles (section 2.3) presented BEFORE their “fabrication” (section 2.4)?

Answer) Thanks for your kind comment. We changed them for fabrication section (2.3) and characterization section (2.4)

2.3. Fabrication of Ce6tetraHA/Ce6decaHA Nanophotosensitizers

20 mg Ce6tetraHA or Ce6decaHA conjugates dissolved in 5 ml water/DMSO mixtures (1/4, v/v) were put into dialysis membrane (MWCO: 8,000 g/mol) and then dialyzed for 1 day with exchange of water every 2~3 h intervals. Dialyzed solution was for analysis or drug release study.

Ce6 contents: 5 mg Ce6tetraHA or Ce6decaHA nanophotosensitizers were reconstituted in 50 ml PBS in the presence of 20mM DTT and then stirred for 24 h. One ml of this solution was diluted with DMSO ten times and, then, Ce6 concentration was evaluated with an Infinite M200 pro microplate reader (Tecan, Mannedorf, Switzerland) at excitation wavelength of 407nm and emission wavelength of 664 nm. For standard test, Ce6 alone in DMSO was diluted 20 times with PBS (20mM DTT) and then diluted ten times with DMSO.

Ce6 content (wt-%) = (Ce6 weight/total weight of nanophotosensitizers)×100.

Ce6 contents in Ce6tetraHA nanophotosensitizers and Ce6decaHA nanophotosensitizers were 22.1 % (w/w) and 39.7 % (w/w), respectively.

2.4. Characterization of Ce6tetraHA or Ce6decaHA Conjugates and Nanophotosensitizers

Synthesis procedures of Ce6tetraHA/Ce6decaHA conjugates was monitored with 1H NMR spectra (500 mHz superconducting Fourier transform (FT)-NMR spectrometer, Varian Unity Inova 500 MHz NB High Resolution FT NMR; Varian Inc, Santa Clara, CA).

Particle size of Ce6tetraHA or Ce6decaHA nanophotosensitizers (concentration: 0.1 %, w/w) was analyzed with Zetasizer Nano-ZS (Malvern, Worcestershire, UK). To estimate average particle size, they were measured particle sizes at least three times and expressed as average±S.D..

Morphology of Ce6tetraHA nanophotosensitizers was observed with transmission electron microscope (TEM) (H-7600, Hitachi Instruments Ltd., Tokyo, Japan). Aqueous solution of nanophotosensitizer was put onto the carbon film coated grid and dried in room temperature for 3h followed by negative stainiung with phosphotungstic acid (0.1%, w/w in deionized water). The observation of nanophotosensitizer morphology was performed at 80kV.

9- Section 2.7: Please, specify the source of the NIH3T3 cells used and describe how they were grown.

Answer) Thanks for your comment. According to your comment, we revised the manuscript and indicated it in experimental section.

NIH3T3 cells and U87MG cells were cultured in DMEM (Gibco, Grand Island, NY, USA) supplemented with 10% (v/v) heat-inactivated fetal bovine serum (FBS) (Invitrogen) and 1% (v/v) antibiotics at 37°C in a 5% CO2 incubator.

10- Legend to Figure 5: (c) does NOT describe “the effect of HAse/DTT”. Also, the error bars in this and other figures suggests multiple experiments. If this is the case, please specify the number of experiments performed.

Answer) Thanks for your comment. Practically, all experiments were average±SD of three (or eight : cell culture experiment) different experiments. We indicated it in each legend of Figures.

Figure 5. The effect of DTT on the Ce6 release dynamics from Ce6tetraHA (a) and Ce6decaHA (b) nanophotosensitizers. (c) The effect of HAse/DTT on the Ce6 release dynamics from Ce6tetraHA nanophotosensitizers. All values are average ± S.D. from three different experiments.

Figure 6. Dark toxicity of Ce6, Ce6tetraHA and Ce6decaHA nanophotosensitizers against NIH3T3mouse fibroblast, HCT116 human colon carcinoma and U87MG human malignant glioma cells. All values are average ± S.D. from results of eight wells.

Figure 7. Ce6 uptake ratio (a) and fluorescence images (b) in the HCT116 cells and U87MG cells. For fluorescence images, Ce6 concentration was adjusted in 3.35 mM (2.0 mg/ml) and exposed to cells for 90 min. All values are average ± S.D. from results of eight wells.

Figure 8. The effect of Ce6 or Ce6tetraHA and Ce6decaHA nanophotosensitizer treatment against cancer cells. ROS generation (a) and PDT treatment. All values are average ± S.D. from results of eight wells.

Figure 9. (a) The effect of blocking of CD44 receptor. Fluorescence images of U87MG cells. For blocking of CD44 receptor, 10 times higher amount of free HA was pre-treated for 60 min before treatment of Ce6 or nanophotosensitizers. (b) ROS generation and PDT effect on the cancer cells. For fluorescence images, Ce6 concentration was adjusted to 3.35 mM (2.0 mg/ml) and exposed to cells for 60 min. All values are average ± S.D. from results of eight wells. *,**; p < 0.001.

11- Figure 5 shows a significant release of Ce6 from the nanoparticles, even in the absence of DTT, GSH or HAse? Can this instability compromise the use of these particles in the clinic? I.e., do they need to be prepared immediately before administration? How feasible is this?

Answer) Thanks for your comment. In fact, this study is preliminary/basic study of novel photosensitizers in vitro and in vivo. They should be investigated more to be improved in stability issues and in vivo anticancer efficacy. If we use them for clinical application in the future, we will be prepared them as a nanophotosensitizer. They will be fabricated as a nanoparticle and prepared for administration. I mean that nanophotosenstizers of mine will be immediately reconstituted in aqueous solution just before administration. This is feasible because HA (a water-soluble segment of nanophotosensitizer) is oriented to the outside of the nanophotosensitizers and it will enable us to prepare aqueous solution.

12- Why were normal cells NOT employed in the PDT studies? It would have been important, as one of the main limitations of PDT is photosensitization. Please, provide data. The same applies to the uptake experiments.

Answer) Thanks for your comment. In fact, photosensitizers do not have specificity against cancer cells and then they can be internalized into normal cells and tissues as same as tumor cells. However, they have no toxicity both of normal cells and tumor cells in the absence of light. ROS (only ROS kill the tumor cells) can be produced by photosensitizers only in the presence of light. Photosensitizers do not produce in the absence of light. This is the reason why patient with PDT should be shaded for a while. Practically, in clinical (or animal study) situations, light must be irradiated to the specific region of body site (in this case, tumor site) and ROS can be produced in the site of light irradiation. From these reason, PDT of normal cells was not applicable. Anyway, we studied Ce6 uptake experiment with normal cells. Practically, Ce6 and nanophotosensitizers also can be internalized into normal cells as well as cancer cells. However, ROS was not produced in the dark condition.

In fact, we are going to try in vivo antitumor study at this moment and we will report them in the separated article near future. Toxicity, PDT effect, ROS generation in the dark condition against normal cells also will be report in that report. Anyway, according to your comment, we showed some results with normal cells as follows: Ce6 and nanophotosensitizers were also internalized into normal cells NIH3T3 cells and RAW264.7 mouse macrophage cells as well as cancer cells but ROS generation in the absence of light irradiation was negligible. Thanks again. We will report these results in the other articles.

13- In the in vitro studies, Ce6 was used for comparison purposes. Why was it not done in the in vivo studies?

Answer) Thanks for your comment. In fact, we are going to do in vivo antitumor study using Ce6tetraHA and Ce6decaHA nanophotosensitizers at this moment. We will report them in the other report in the future. Thanks.

Other issues:

- The manuscript contains several awkward sentences that should be rephrased, to improve the manuscript’s readability. Below are some examples:

* lns 64-65: “and then non-toxic reaction is expected in the normal tissues or cells [5-7]”.

Answer) Thanks to your kind comment. According to your comment, we revised the manuscript.

When tumor tissues are irradiated with appropriate wavelength of light, photosensitizers produce ROS in tumor tissues only but not in the normal tissues or cells [5-7]. Then, photosensitizers kill the cancer cells by over-production of ROS in tumor tissues.

* lns 72-73: “Ce6-polyvinylpyrrolidone (PVP) formulations revealed less in vivo intrinsic phototoxicity”. This sentence is confusing. A lower phototoxicity should not be viewed as an advantage in the context of PDT.

Answer) Thanks to your kind comment. According to your comment, we revised the manuscript. To avoid confusing, we removed the sentence.

For example, Chin et al., reported that Ce6-polyvinylpyrrolidone (PVP) formulations revealed faster clearance rate from the body compared to Ce6 alone [13].

* lns 75-76: “nanophotosensitizers composed of poly(ethylene glycol) (PEG)-Ce6 conjugates showed activated state in aqueous solution”. What is the meaning of “showed activated state”?

Answer) Thanks to your kind comment. According to your comment, we revised the manuscript.

We previously reported that nanophotosensitizers composed of poly(ethylene glycol) (PEG)-Ce6 conjugates showed superior aqueous solubility, higher fluorescence intensity in aqueous solution, higher photodynamic efficacy against HCT116 cells in vitro and in vivo tumor models compared to Ce6 alone [15].

* lns 81-82: “Nanomedicine-based photodynamic therapy also enables disease diagnosis and image-guided surgery [19-21].” How can PDT be used in diagnosis and surgery?

Answer) Thanks to your kind comment. In fact, Photosensitizers are strong fluorescent dye. Then those have been investigated as a diagnostic agent instead of radiolabelling agent. For example, photosensitizer can be administered to tumor patients and the region of tumor (normally in the case of gastrointestinal cancer) can be confirmed with PET-CT (or other kind of fluorescence imaging equipment) and then tumors can be removed by surgery.

* ln 89: What is the meaning of higher redox state?

Answer) Thanks to your kind comment. In fact, the reduction/oxidation status of cancer cells is normally very different according to the types of cancer cells and/or origin of cancer c cells. It means that activity of reduction/oxidation is elevated in cancer cells compared to normal cells/tissues.

On the other hand, abnormal redox status of tumor microenvironment is also spotlighted in recent decades since high metabolic activity in tumor tissues promotes reduction/oxidation (redox) activities [26-30]. Many scientists have been applied abnormal redox status of tumor as a therapeutic target for stimuli-sensitive drug delivery [26-30].

* lns 87-89: “Compared to normal cells and tissues, tumor cells express various receptors with higher degree, higher enzymatic activity such as hyaluronidase (HAse), unbalanced immune system, poor perfusion, acidic pH environment and higher redox-status [22-24].” It is important to distinguish the tumor cells from the tumor microenvironment. Tumor cells do NOT express and unbalanced immune system.

Answer) Thanks to your kind comment. According to your comment, we revised the manuscript. To avoid confused sentences, we removed some sentences and revised.

Compared to normal cells and tissues, tumor cells are characterized by higher expression of receptors, higher enzymatic activity such as hyaluronidase (HAse), poor perfusion, acidic pH environment and higher redox-status compared to normal cells [22-24].

* lns 97-98: What is the meaning of “high metabolic activity in tumor tissues promote reduction/oxidation activities”?

Answer) Thanks to your kind comment. According to your comment, we revised the manuscript. These sentences mean that high GSH levels and ROS levels are high in the tumor tissues compared to normal cells and tissues. And these metabolites promote reduction (GSH) and oxidation (ROS) activities.

* ln 107-108: The sentence “HA has hydrophilic macromolecules” is confusing.

Answer) Thanks to your kind comment. According to your comment, we revised the manuscript.

Furfurthermor, Ce6tetraHA or Ce6decaHA conjugates must be formed nanoscale vehicles as nanophotosensitizers since Ce6 itself is a hydrophobic molecule while HA is a hydrophilic macromolecule.

* ln 244: “Ce6 or nanophotosensitizers were exposed to cells”. It is the cells that are exposed to the nanophotosensitizers, not the other way round.

Answer) Thanks to your kind comment. According to your comment, we revised the manuscript. To avoid confusing, this sentence was changed as follows:

Cells (2×104 cells) in a 96-well plate were treated with Ce6 alone or nanophotosensitizers.

* lns 263-264: “When the solid tumor were reached larger than 4 mm in diameter”. This sentence is confusing.

Answer) Thanks to your kind comment. According to your comment, we revised the manuscript. This sentence means that the diameter of tumor became larger than 4 mm ~. Anyway we revised the manuscript to avoid confusing.

When diameter of solid tumor became larger than 4 mm, Ce6tetraHA nanophotosensitizer solution (10 mg/kg) was administered intravenously (i.v.) via tail vein of the mice. The volume of injection solution was 100 ml.

* lns 386-389: The sentences “Figure 7 showed the comparison of Ce6 and nanophotosensitizers in the Ce6 uptake ratio in HCT116 cells and U87MG cells in vitro.” and “all chemicals showed that Ce6 uptake ratio at HCT116 cells and U87MG cells was gradually increased according to the increase of Ce6 concentration.” are very confusing.

Answer) Thanks to your kind comment. According to your comment, we revised the manuscript.

Figure 7 shows the Ce6 uptake ratio of Ce6 alone or nanophotosensitizers in cancer cells in vitro. As shown in Figure 7(a), Ce6 uptake ratio in cancer cells was gradually increased at all formulations with dose dependently.

* lns 401-402: Please, improve the sentence “differences in ROS generation and PDT efficacy was practically small between Ce6tetraHA and Ce6decaHA nanophotosensitizers”.

Answer) Thanks to your kind comment. According to your comment, we revised the manuscript.

Also, Ce6tetraHA and Ce6decaHA nanophotosensitizers have similar efficacy in ROS generation and PDT.

* ln408: “HA was pretreated to cells”. Please, improve.

Answer) Thanks to your kind comment. According to your comment, we revised the manuscript.

CD44 receptor was blocked by pretreatment with free HA to evaluate CD44 receptor targetability of Ce6tetraHA nanophotosensitizer.

* lns 412-413: “Ce6tetraHA nanophotosensitizer can be delivered to the cells through CD44 receptor”. This sentence suggests that molecules may enter cells through receptors, whereas they do so through transporters.

Answer) Thanks to your kind comment. According to your comment, we revised the manuscript. In this sentence, we used molecules may enter cells through receptors.

* lns 479-480: “Ce6tetraHA nanophotosensitizers displayed redox-sensitive and HAse-sensitive behavior in photosensitizer release”. This sentence is not very clear.

Answer) Thanks to your kind comment. According to your comment, we revised the manuscript.

Furthermore, nanophotosensitizers can be delivered to cancer cells by redox-sensitivity and HAse-sensitivity. Then, Ce6 release from nanophotosensitizers is controlled by DTT (redox sensitive) and HAse in tumor cells.

* lns 485-486: “this peculiarity have been considered as a cancer target of bioactive molecules [28,50,51]”: Confusing.

Answer) Thanks to your kind comment. According to your comment, we revised the manuscript.

Especially, intracellular GSH level in cancer cells is significantly higher than normal cells and has been considered as a cancer target [27,28,50-52].

* lns 488-490: “we approved that Ce6 can be activated through degradation of disulfide bond by GSH treatment to nanoparticles and then accelerated drug release with improved cancer cell toxicity [28].” Very confusing. For instance, what is the meaning of “activated” in this context?

Answer) Thanks to your kind comment. According to your comment, we revised the manuscript. To avoid confusing, we changed the sentences as follows.

In previous report, we approved that fluorescence intensity of Ce6 was increased by degradation of disulfide bond in the nanoparticles and then release of anticancer drug was accelerated [28].

- It is advisable to use the use the 'spell-check' and 'grammar-check' functions of the word processor, to minimize spelling and grammar (namely in terms of verb conjugation) errors.

Answer) Thanks to your kind comment. According to your comment, we revised the manuscript. We checked spelling and grammar.

Reviewer 3 Report

This manuscript describes the development of new nanophotosensitizers-conjugates for photodynamic therapy consisting of hyaluronic acid (HA) targeting cancer cells with high CD44 receptors expression and hyperbranched chlorin e6 (Ce6) containing redox‐sensitive disulfide linkages. PDT efficiency and receptor-mediated cellular uptake of conjugates, Ce6tetraHA and Ce6decaHA, in the cells were demonstrated, compared to free Ce6. The in vivo studies also confirmed their high selectivity for tumors. Furthermore, the authors have shown the redox-sensitive and Hyaluronidase-sensitive features of these conjugates in solution. This is an interesting study in the field of Targeted Photodynamic Therapy of Cancer.

However, some issues need to be addressed.

General:

The manuscript has grammar and syntax problems and, therefore, needs to be revised.

In my opinion, the choice of the terminology “conjugates” for all structures is not particularly appropriate and paper gets a bit confusing. It should only be used for Ce6 tetramer and decamer conjugated with hyaluronic acid. Therefore, it would be worth trying to make it clearer (e.g. expressions like “functionalized or functionalization” should be used).

I suggest the authors reformulate the Abstract taking into account the previous considerations.

The introduction is well written, but it would be interesting have a short paragraph with few examples of Hyaluronic acid-based nanoparticles for targeted PDT (Biomaterials 33 (2012) 3980-3989; ACS Appl. Mater. Interfaces, 9 (2017) 32509-32519).

Throughout the manuscript, some errors should be corrected concerning the units (e.g. space should be added between the value and the unit).

The poor quality of the 1H NMR spectra in Figures 1 and 2 and the lack of additional characterization data is a problem for demonstrating that the proposed structures were reached. I recommend the authors to provide full 1H NMR spectra, with integrated peaks, and high-resolution mass spectra (HRMS) at least of the precursors of conjugates, Ce6 tetramer and Ce6 decamer, as supporting material. 

The number of independent experiments should be mentioned in cell studies.

Specific Remarks:

Line 39: Replace “carboxylic” with “carboxyl”.

Line 64: “photosensitizer produces ROS excessively in tumor tissues”. What do the authors consider excessively?

Lines 65-67: “Ce6 has been investigated extensively for PDT of cancer because it can be activated at higher wavelength of light and deeper penetration efficiency than traditional photosensitizers”. Please be more specific regarding traditional photosensitizers.

Lines 67-69: The statement “most of photosensitizers including Ce6 has no specificity for cancer cells, low penetration depth in the tumor tissues and necessity for long-term sun-shade” presents some contradiction with “safe option for treatment of cancer” (line 61).

Lines 75-77: “We previously reported that nanophotosensitizers composed of poly(ethylene glycol) (PEG)-Ce6 conjugates showed activated state in aqueous solution, higher photodynamic efficacy against  HCT116 cells in vitro and in vivo tumor model” Please clarify “showed activated state” and the comparison “higher photodynamic efficacy” is incomplete.

Lines 79-80:  “higher PDT efficacy”. Incomplete comparison.

Lines 93-95: “Lee and Jeong reported that nanoparticles of HA-Ce6-93 poly(L-histidine) conjugates specifically deliver the anticancer drug to a CD44 receptor of cancer cells 94 at in vitro and in vivo”. Cite the works here.

Line 109: Replace “must be fully degraded” with “should be degraded”.

Lines 130-131: Replace “obtained Ce6 - triple cystamine (Ce6-(-NSSNH2)3) conjugates” with “Ce6 functionalized with three cystamine moieties was obtained”.

Lines 133-134: Replace “98mg of Ce6-(-NSSNH2)3 133 conjugates” with “Ce6-(-NSSNH2)3 (98 mg)”.

Lines: 131, 137, 145, 152, 163, 170, 177: More details about the obtained products (solid or oil, quantities).

Line 180: Please remove “procedures” and clarify “monitored”.

Line 190: Replace “Fabrication” with “Preparation”.

Line 191: Replace “20 mg Ce6tetraHA or Ce6decaHA conjugates” with “Ce6tetraHA or Ce6decaHA conjugates (20 mg)”.

Line 194: Replace “5 mg Ce6tetraHA or Ce6decaHA nanophotosensitizers” with “Ce6tetraHA or Ce6decaHA nanophotosensitizers” (5 mg).

Line 264: Clarify the concentration of photosensitizer administered.

Lines 276-307:  The authors state that “specific peaks of Ce6 and cystamine” confirms the formation of the claimed conjugates. Further details of spectra should be provided (see general remarks). The captions for NMR spectra must contain solvent and frequency.

Lines 306 and 309: Remove “scheme”

Line 407: (b) is missing. Detailed conditions of PDT experiments should be described (light dose, wavelength).

Author Response

Answer to Reviewer 3’s comment

This manuscript describes the development of new nanophotosensitizers-conjugates for photodynamic therapy consisting of hyaluronic acid (HA) targeting cancer cells with high CD44 receptors expression and hyperbranched chlorin e6 (Ce6) containing redox‐sensitive disulfide linkages. PDT efficiency and receptor-mediated cellular uptake of conjugates, Ce6tetraHA and Ce6decaHA, in the cells were demonstrated, compared to free Ce6. The in vivo studies also confirmed their high selectivity for tumors. Furthermore, the authors have shown the redox-sensitive and Hyaluronidase-sensitive features of these conjugates in solution. This is an interesting study in the field of Targeted Photodynamic Therapy of Cancer.

However, some issues need to be addressed.

General:

The manuscript has grammar and syntax problems and, therefore, needs to be revised.

Answer) According to your comment, we revised the manuscript and corrected the English expression.

In my opinion, the choice of the terminology “conjugates” for all structures is not particularly appropriate and paper gets a bit confusing. It should only be used for Ce6 tetramer and decamer conjugated with hyaluronic acid. Therefore, it would be worth trying to make it clearer (e.g. expressions like “functionalized or functionalization” should be used).

Answer) Thanks for your comment. According to your comment, We revised the manuscript. We used the term “Conjugates” for Ce6 tetramer and decamer conjugated with hyaluronic acid.

I suggest the authors reformulate the Abstract taking into account the previous considerations.

Answer) According to your comment, we revised the manuscript and added some sentences. Furthermore, previous study ours are introduced in the introduction part and discussion part.

Abstract: The aim of this study is to synthesize novel types of nanophotosensitizers based on hyperbranched chlorin e6 (Ce6)-conjugated hyaluronic acid (HA) for redox-sensitive and CD44-sensitive photodynamic therapy (PDT) of cancer cells. Hyperbranched Ce6 was synthesized by conjugation of Ce6 each other using disulfide linkage. Since most of the previous studies regarding nanophotosensizers are concerned with simple conjugation between monomeric units of photosensitizer and polymer materials, hyperbranched Ce6 was considered to make novel types of macromolecular photosensitizer. were conjugated with cystamine and then amine end group of Ce6-cystamine was conjugated again with three equivalents of Ce6 to make Ce6 tetramer. Ce6 decamer as a hyperbranched Ce6 was synthesized by conjugation of six equivalents of Ce6 to Ce6 tetramer using disulfide linkage. HA-cystamine was conjugated with Ce6 tetramer or Ce6 decamer to make HA-Ce6 tetramer (Ce6tetraHA) or HA-Ce6 decamer (Ce6decaHA) conjugates. Ce6tetraHA and Ce6decaHA nanophotosensitizers have small particle sizes less than 200 nm with spherical shapes. In vitro drug release study showed that Ce6 release was increased in the presence of dithiothreitol (DTT) and hyaluronidase (HAse). These results indicated that Ce6tetraHA nanophotosensitizers have redox-sensitive and HAse-sensitive release properties. Furthermore, Ce6tetraHA nanophotosensitizers was resulted in higher intracellular Ce6 accumulation, higher ROS generation and higher PDT efficacy than that of Ce6 alone. Due to the HA, Ce6tetraHA nanophotosensitizers were delivered by CD44 receptor-mediated manner in vitro cell culture study and in vivo tumorxenograft study, i.e. fluorescence intensity in the tumor tissues was significantly stronger than those of other organs. CD44 receptor blocking by HA pretreatment induced that fluorescence intensity in tumor tissues was lower than that of liver, indicating that Ce6tetraHA nanophotosensitizers can be delivered to tumors by redox-sensitive and CD44-sensitive manner. We suggest Ce6tetraHA and Ce6decaHA nanophotosensitizers as a promising candidate for PDT of cancers.

The introduction is well written, but it would be interesting have a short paragraph with few examples of Hyaluronic acid-based nanoparticles for targeted PDT (Biomaterials 33 (2012) 3980-3989; ACS Appl. Mater. Interfaces, 9 (2017) 32509-32519).

Answer) Thanks for your comment. According to your comment, we revised the manuscript and introduced more about their studies.

Yoon et al., reported that Ce6-conjugated HA nanoparticles can be delivered tumor-specific enzyme such as HAse and CD44-receptor-mediated pathway [22]. Gao et al., reported that Ce6-encapsulated HA nanoparticles were efficiently accumulated in tumor xenograft of human colon cancer and observed no apparent systemic toxicity against mice [23]. And we added these references.

Yoon, H.Y.; Koo, H.; Choi, K.Y.; Lee, S.J.; Kim, K.; Kwon, I.C.; Leary, J.F.; Park, K.; Yuk, S.H.; Park, J.H.; Choi, K. Tumor-targeting hyaluronic acid nanoparticles for photodynamic imaging and therapy. Biomaterials. 2012, 33, 3980-3989.

Gao, S.;, Wang, J.; Tian, R.; Wang, G.; Zhang, L.; Li, Y.; Li, L.; Ma, Q.; Zhu, L. Construction and Evaluation of a Targeted Hyaluronic Acid Nanoparticle/Photosensitizer Complex for Cancer Photodynamic Therapy. ACS Appl. Mater. Interfaces. 2017, 9, 32509-32519.

Throughout the manuscript, some errors should be corrected concerning the units (e.g. space should be added between the value and the unit).

Answer) Thanks for your comment. According to your comment, we revised the manuscript and corrected concerning the units.

The poor quality of the 1H NMR spectra in Figures 1 and 2 and the lack of additional characterization data is a problem for demonstrating that the proposed structures were reached. I recommend the authors to provide full 1H NMR spectra, with integrated peaks, and high-resolution mass spectra (HRMS) at least of the precursors of conjugates, Ce6 tetramer and Ce6 decamer, as supporting material. 

Answer) Thanks for your comment. At this moment, we couldn’t obtain high resolution of NMR spectra because hyperbranched Ce6 tetramer and Ce6 decamer has complex structure. We will report the precise characterized of Ce6 tetramer and Ce6 decamer in a separated report. However, we added NMR spectra of Ce6 tetramer and Ce6 decamer as supporting material. Moreover, we added Ce6 contents in nanophotosensitizers in Table 1 to support further characterization.

Figure s1. 1H NMR spectra of Ce6 tetramer (b) and Ce6 decamer (c). For measurement of 1H NMR (500 mHz) spectra, Ce6 tetramer and Ce6 decamer were dissolved in deuterated DMSO (DMSO d6).

The number of independent experiments should be mentioned in cell studies.

Answer) Thanks for your comment. According to your comment, we indicated the number of independent experiments for cell studies in Figure captions.

Figure 6. Dark toxicity of Ce6, Ce6tetraHA and Ce6decaHA nanophotosensitizers against NIH3T3mouse fibroblast, HCT116 human colon carcinoma and U87MG human malignant glioma cells. All values are average ± S.D. from results of eight wells.

Figure 7. Ce6 uptake ratio (a) and fluorescence images (b) in the HCT116 cells and U87MG cells. For fluorescence images, Ce6 concentration was adjusted in 3.35 mM (2.0 mg/ml) and exposed to cells for 90 min. All values are average ± S.D. from results of eight wells.

Figure 8. The effect of Ce6 or Ce6tetraHA and Ce6decaHA nanophotosensitizer treatment against cancer cells. ROS generation (a) and PDT treatment (b). Cells were irradiated at 664 nm (2 J/cm2). All values are average ± S.D. from results of eight wells.

Figure 9. (a) The effect of blocking of CD44 receptor. Fluorescence images of U87MG cells. For blocking of CD44 receptor, 10 times higher amount of free HA was pre-treated for 60 min before treatment of Ce6 or nanophotosensitizers. (b) ROS generation and PDT effect on the cancer cells. For fluorescence images, Ce6 concentration was adjusted to 3.35 mM (2.0 mg/ml) and exposed to cells for 60 min. All values are average ± S.D. from results of eight wells. *,**; p < 0.001.

Specific Remarks:

Line 39: Replace “carboxylic” with “carboxyl”.

Answer) Thanks for your comment. According to your comment, We revised the manuscript

Carboxyl groups of Ce6 were conjugated with cystamine and then amine end group of Ce6-cystamine was conjugated again with three equivalents of Ce6 to make Ce6 tetramer.

Line 64: “photosensitizer produces ROS excessively in tumor tissues”. What do the authors consider excessively?

Answer) Thanks for your comment. Normally, cancer cells and tumor tissues have higher content of ROS compared to normal cells. Elevated levels of ROS in tumor tissues induce proliferation, migration, invasion and metastasis of cancer cells. However, apoptosis or necrosis of cancer cells can be induced if ROS level in cancer cells exceeds certain level. “Excessively” means these properties in cancer cells.

Lines 65-67: “Ce6 has been investigated extensively for PDT of cancer because it can be activated at higher wavelength of light and deeper penetration efficiency than traditional photosensitizers”. Please be more specific regarding traditional photosensitizers.

Answer) Thanks for your comments. We revised he manuscript according to your comment.

Ce6 has been extensively investigated for PDT of cancer because it has deeper tissue penetration efficiency and is activated at higher wavelength of light than traditional photosensitizers such as 5-aminolevulic acid (5-ALA) or porphyrins, [8,9].

Lines 67-69: The statement “most of photosensitizers including Ce6 has no specificity for cancer cells, low penetration depth in the tumor tissues and necessity for long-term sun-shade” presents some contradiction with “safe option for treatment of cancer” (line 61).

Answer) Thanks for your comments. We revised he manuscript according to your comment. These sentences means that therapeutic efficacy of Ce6 is still insufficient against cancer patients. However, “safe option for treatment of cancer” means that PDT has little undesirable side effects compared to traditional chemotherapy. Anyway, we revised the manuscript to avoid contradiction of this sentence.

Photodynamic therapy (PDT) is generally composed of non-toxic components such as light, photosensitizer and oxygen [1]. PDT is believed to be a safe treatment option for cancer patients since it has little undesirable side-effects compared to traditional chemotherapy [1-3].

Despite of these advantages, most of photosensitizers including Ce6 has still low cancer cell specificity and low delivery capacity against the tumor tissues and requirement for long-term sun-shade [3,9-11].

Lines 75-77: “We previously reported that nanophotosensitizers composed of poly(ethylene glycol) (PEG)-Ce6 conjugates showed activated state in aqueous solution, higher photodynamic efficacy against  HCT116 cells in vitro and in vivo tumor model” Please clarify “showed activated state” and the comparison “higher photodynamic efficacy” is incomplete.

Answer) Thanks for your comments. We revised he manuscript according to your comment.

We previously reported that nanophotosensitizers composed of poly(ethylene glycol) (PEG)-Ce6 conjugates showed superior aqueous solubility, higher fluorescence intensity in aqueous solution, higher PDT efficacy against HCT116 cells in vitro and in vivo tumor models compared to Ce6 alone [15].

Lines 79-80:  “higher PDT efficacy”. Incomplete comparison.

Answer) Thanks for your comments. We revised he manuscript according to your comment.

Furthermore, nanophotosensitizer of chitosan-Ce6 complexes has higher PDT efficacy against cholangiocarcinoma cells and higher absorption efficiency in bile duct tissue explants compared to Ce6 alone

Lines 93-95: “Lee and Jeong reported that nanoparticles of HA-Ce6-93 poly(L-histidine) conjugates specifically deliver the anticancer drug to a CD44 receptor of cancer cells 94 at in vitro and in vivo”. Cite the works here.

Answer) Thanks for your comments. We revised he manuscript according to your comment.

Lee and Jeong reported that nanoparticles of HA-Ce6-poly(L-histidine) conjugates specifically deliver the anticancer drug to a CD44 receptor of cancer cells at in vitro and in vivo [28].

Line 109: Replace “must be fully degraded” with “should be degraded”.

Answer) Thanks for your comments. We revised he manuscript according to your comment.

Ce6tetraHA or Ce6decaHA conjugates should be degraded in tumor cells by glutathione (GSH)

Lines 130-131: Replace “obtained Ce6 - triple cystamine (Ce6-(-NSSNH2)3) conjugates” with “Ce6 functionalized with three cystamine moieties was obtained”.

Answer) Thanks for your comments. We revised he manuscript according to your comment.

Dialyzed solution was freeze-dried for 3 days and Ce6 functionalized with three cystamine moieties was obtained (Ce6-(-NSSNH2)3).

Lines 133-134: Replace “98mg of Ce6-(-NSSNH2)3 133 conjugates” with “Ce6-(-NSSNH2)3 (98 mg)”.

Answer) Thanks for your comments. We revised he manuscript according to your comment.

then mixed with Ce6-(-NSSNH2)3 (98mg)

Lines: 131, 137, 145, 152, 163, 170, 177: More details about the obtained products (solid or oil, quantities).

Answer) Thanks for your comments. We revised he manuscript according to your comment.

Dialyzed solution was freeze-dried for 3 days and Ce6 functionalized with three cystamine moieties was obtained as a solid (Ce6-(-NSSNH2)3).

This solution was lyophilized for 3 days and then Ce6 tetramer was obtained as a solid.

Then, Ce6 tetramer functionalized with six cystamine moieties was obtained as a sold and this was was refrigerated until used.

Then, Ce6 decamer was obtained as a solid.

This was lyophilized for 2 days to obtain HA-cystamine as a solid.

Following this, resulting solution was lyophilized for 3 days to obtain Ce6tetraHA conjugates as a solid.

This was dialyzed against water over 2 days and lyophilized for 3 days to obtain Ce6decaHA conjugates as a solid.

Line 180: Please remove “procedures” and clarify “monitored”.

Answer) Thanks for your comments. We revised he manuscript according to your comment.

Synthesis of Ce6tetraHA/Ce6decaHA conjugates was monitored with 1H NMR spectra (500 mHz superconducting Fourier transform (FT)-NMR spectrometer, Varian Unity Inova 500 MHz NB High Resolution FT NMR; Varian Inc, Santa Clara, CA).

Synthesis of Ce6tetraHA/Ce6decaHA conjugates was confirmed with 1H NMR spectra

Line 190: Replace “Fabrication” with “Preparation”.

Answer) Thanks for your comments. We revised he manuscript according to your comment.

2.3. Preparation of Ce6tetraHA/Ce6decaHA Nanophotosensitizers

Line 191: Replace “20 mg Ce6tetraHA or Ce6decaHA conjugates” with “Ce6tetraHA or Ce6decaHA conjugates (20 mg)”.

Answer) Thanks for your comments. We revised he manuscript according to your comment.

Ce6tetraHA or Ce6decaHA conjugates (20 mg) dissolved in 5 ml water/DMSO mixtures (1/4, v/v) were put into dialysis membrane (MWCO: 8,000 g/mol) and then dialyzed for 1 day with exchange of water every 2~3 h intervals.

Line 194: Replace “5 mg Ce6tetraHA or Ce6decaHA nanophotosensitizers” with “Ce6tetraHA or Ce6decaHA nanophotosensitizers” (5 mg).

Answer) Thanks for your comments. We revised he manuscript according to your comment.

Ce6tetraHA or Ce6decaHA nanophotosensitizers (5 mg) were reconstituted in 50 ml PBS in the presence of 20mM DTT and then stirred for 24 h.

Line 264: Clarify the concentration of photosensitizer administered.

Answer) Thanks for your comments. We revised he manuscript according to your comment.

When diameter of solid tumor became larger than 4 mm, Ce6tetraHA nanophotosensitizer solution (5.0 mg/kg) was administered intravenously (i.v.) via tail vein of the mice. The volume of injection solution was 100 ml.

Lines 276-307:  The authors state that “specific peaks of Ce6 and cystamine” confirms the formation of the claimed conjugates. Further details of spectra should be provided (see general remarks). The captions for NMR spectra must contain solvent and frequency.

Answer) Thanks for your comments. We revised he manuscript according to your comment.

Figure 1. Synthesis and 1H NMR spectra of Ce6-cystamine (a), Ce6 tetramer (b) and Ce6 decamer (c). For measurement of 1H NMR (500 mHz) spectra, Ce6, Ce6 tetramer and Ce6 decamer were dissolved in deuterated DMSO (DMSO d6).

Figure 2. Synthesis and 1H NMR spectra of (a) Ce6tetraHA conjugates and (b) Ce6decaHA conjugates. For measurement of 1H NMR (500 mHz) spectra, Ce6tetraHA and Ce6decaHA conjugates were distributed in 0.2 ml of D2O and then mixed with 0.8 ml DMSO-d6.

Lines 306 and 309: Remove “scheme”

Answer) Thanks for your comments. We revised he manuscript according to your comment.

Figure 1. Synthesis and 1H NMR spectra of Ce6-cystamine (a), Ce6 tetramer (b) and Ce6 decamer (c). For measurement of 1H NMR (500 mHz) spectra, Ce6, Ce6 tetramer and Ce6 decamer were dissolved in deuterated DMSO (DMSO d6).

Figure 2. Synthesis and 1H NMR spectra of (a) Ce6tetraHA conjugates and (b) Ce6decaHA conjugates. For measurement of 1H NMR (500 mHz) spectra, Ce6tetraHA and Ce6decaHA conjugates were distributed in 0.2 ml of D2O and then mixed with 0.8 ml DMSO-d6.

Line 407: (b) is missing. Detailed conditions of PDT experiments should be described (light dose, wavelength).

Answer)

Figure 8. The effect of Ce6 or Ce6tetraHA and Ce6decaHA nanophotosensitizer treatment against cancer cells. ROS generation (a) and PDT treatment (b). Cells were irradiated at 664 nm (2 J/cm2). All values are average ± S.D. from results of eight wells.

Round 2

Reviewer 2 Report

This is an improved version of the original manuscript. However, there are a few issues that still need to be addressed. Please, see below, after the authors’ replies to this reviewers’ original comments.

Answer to Reviewer 2’s comment

1- Title: In what sense are the new materials biodegradable? And how was this “biodegradability” assessed?

Answer) Thanks for your comments. We used “biodegradable” in the title because HA can be degradable by hyaluronidase and Ce6 tetramer/Ce6decamer can be divided to Ce6 monomer by GSH or DTT. Then all components of Ce6tetraHA NP or Ce6decaHA NP can be degraded to monomeric units. Then we used “biodegradable.

The term “biodegradable” is usually used in the context of pollution. It seems to be out of the present study context and should preferentially be removed.

2- lns 39-40: What is the meaning of “redox-sensitive and CD44-sensitive photodynamic therapy”? Figure 5 shows that DTT, GSH and hyaluronidase significantly increase the release of Ce6 from the nanomaterials, but is this release advantageous? The results presented in this manuscript do not suggest so.

Answer) Thanks for your comments. In fact, we used redox-sensitive due to that disulfide linkage can be degraded by DTT or GSH (GSH is an anti-oxidant material, which is abundant in cancer cells.). And this linkage can be disintegrated by GSH in cancer cells. Then we used redox-sensitive because reduction/oxidation potential of cancer cells is significantly higher than normal cells. Furthermore, due to the GSH concentration in cancer cells, Ce6 release rate must be accelerated in the cancer cells because GSH content in the cancer cells are higher than that in normal cells (at least 100 times higher). Anyway, we discussed more in Discussion section and added ref.

For example, redox-potential in cancer cells is higher than normal cells or tissues and GSH is elevated during oxidative stress [50,51]. Especially, intracellular GSH level in cancer cells is significantly higher than normal cells and this peculiarity has been considered as a cancer target of bioactive molecules [27,28,50-52].

Gamcsik, M.P.; Kasibhatla, M.S.; Teeter, S.D.; Colvin, O.M. Glutathione levels in human tumors. Biomarkers. 2012, 17, 671-691.

Bansal, A.; Simon, M.C. Glutathione metabolism in cancer progression and treatment resistance. J. Cell Biol. 2018, 217, 2291-2298.

- The wording “redox-sensitive and CD44-sensitive photodynamic therapy” is confusing.

- The authors did not address the question regarding the release of Ce6 from the nanomaterials.

3- ln 49: “resulted in higher intracellular Ce6 accumulation”. How did the authors discriminate between Ce6 accumulation and Ce6tetraHA or Ce6decaHA accumulation? Also, the two nanoparticles under consideration are abbreviated in several different ways throughout the manuscript (including figures). Please, ensure consistency.

Answer) Thanks for your comments. As shown in Experimental section (2.4. fabrication of Ce6tetraHA/Ce6decaHA nanophotosensitizers), Ce6 contents in Ce6tetraHA NP or Ce6decaHA NP were measured by degradation of nanophotosensitizers with DTT as shown in Experimental section (2.4. fabrication of Ce6tetraHA/Ce6decaHA nanophotosensitizers). Cells were treated with Ce6tetraHA and Ce6decaHA nanophotosensitizers based on Ce6 contents. Anyway, we revised the manuscripts in the Experimental section and Results section (in Table 1).

In Experimental section (2.4. fabrication of Ce6tetraHA/Ce6decaHA nanophotosensitizers).

Ce6 contents in Ce6tetraHA nanophotosensitizers and Ce6decaHA nanophotosensitizers were 22.1 % (w/w) and 39.8 % (w/w), respectively.

- Please, the title of section 2.4 is very confusion and needs to be rephrased. Also, there seems to be some repetition between sections 2.4 and 2.6.

- The issue raised was: Is the experimental design adopted adequate to distinguish the emission of Ce6 from those of the conjugates and nanophotosensitizers? Please, provide emission spectra for the Ce6, as well as for Ce6tetraHA and Ce6decaHA conjugates and nanophotosensitezers at an excitation wavelength of 407 nm.

9- Section 2.7: Please, specify the source of the NIH3T3 cells used and describe how they were grown.

Answer) Thanks for your comment. According to your comment, we revised the manuscript and indicated it in experimental section.

NIH3T3 cells and U87MG cells were cultured in DMEM (Gibco, Grand Island, NY, USA) supplemented with 10% (v/v) heat-inactivated fetal bovine serum (FBS) (Invitrogen) and 1% (v/v) antibiotics at 37°C in a 5% CO2 incubator.

- Please, specify the source of the cell lines used.

- Please, specify the antibiotics added to the culture medium.

10- Legend to Figure 5: (c) does NOT describe “the effect of HAse/DTT”. Also, the error bars in this and other figures suggests multiple experiments. If this is the case, please specify the number of experiments performed.

Answer) Thanks for your comment. Practically, all experiments were average±SD of three (or eight : cell culture experiment) different experiments. We indicated it in each legend of Figures.

Figure 5. The effect of DTT on the Ce6 release dynamics from Ce6tetraHA (a) and Ce6decaHA (b) nanophotosensitizers. (c) The effect of HAse/DTT on the Ce6 release dynamics from Ce6tetraHA nanophotosensitizers. All values are average ± S.D. from three different experiments.

Figure 6. Dark toxicity of Ce6, Ce6tetraHA and Ce6decaHA nanophotosensitizers against NIH3T3mouse fibroblast, HCT116 human colon carcinoma and U87MG human malignant glioma cells. All values are average ± S.D. from results of eight wells.

- If results are from eight wells, it means that they are from a single independent experiment with eight replicate wells for each condition. This must be stated in the manuscript.

Figure 7. Ce6 uptake ratio (a) and fluorescence images (b) in the HCT116 cells and U87MG cells. For fluorescence images, Ce6 concentration was adjusted in 3.35 mM (2.0 mg/ml) and exposed to cells for 90 min. All values are average ± S.D. from results of eight wells.

- If results are from eight wells, it means that they are from a single independent experiment with eight replicate wells for each condition. This must be stated in the manuscript.

Figure 8. The effect of Ce6 or Ce6tetraHA and Ce6decaHA nanophotosensitizer treatment against cancer cells. ROS generation (a) and PDT treatment. All values are average ± S.D. from results of eight wells.

- If results are from eight wells, it means that they are from a single independent experiment with eight replicate wells for each condition. This must be stated in the manuscript.

Figure 9. (a) The effect of blocking of CD44 receptor. Fluorescence images of U87MG cells. For blocking of CD44 receptor, 10 times higher amount of free HA was pre-treated for 60 min before treatment of Ce6 or nanophotosensitizers. (b) ROS generation and PDT effect on the cancer cells. For fluorescence images, Ce6 concentration was adjusted to 3.35 mM (2.0 mg/ml) and exposed to cells for 60 min. All values are average ± S.D. from results of eight wells. *,**; p < 0.001.

- If results are from eight wells, it means that they are from a single independent experiment with eight replicate wells for each condition. This must be stated in the manuscript.

11- Figure 5 shows a significant release of Ce6 from the nanoparticles, even in the absence of DTT, GSH or HAse? Can this instability compromise the use of these particles in the clinic? I.e., do they need to be prepared immediately before administration? How feasible is this?

Answer) Thanks for your comment. In fact, this study is preliminary/basic study of novel photosensitizers in vitro and in vivo. They should be investigated more to be improved in stability issues and in vivo anticancer efficacy. If we use them for clinical application in the future, we will be prepared them as a nanophotosensitizer. They will be fabricated as a nanoparticle and prepared for administration. I mean that nanophotosenstizers of mine will be immediately reconstituted in aqueous solution just before administration. This is feasible because HA (a water-soluble segment of nanophotosensitizer) is oriented to the outside of the nanophotosensitizers and it will enable us to prepare aqueous solution.

- This should be clearly stated in the manuscript.

* lns 81-82: “Nanomedicine-based photodynamic therapy also enables disease diagnosis and image-guided surgery [19-21].” How can PDT be used in diagnosis and surgery?

Answer) Thanks to your kind comment. In fact, Photosensitizers are strong fluorescent dye. Then those have been investigated as a diagnostic agent instead of radiolabelling agent. For example, photosensitizer can be administered to tumor patients and the region of tumor (normally in the case of gastrointestinal cancer) can be confirmed with PET-CT (or other kind of fluorescence imaging equipment) and then tumors can be removed by surgery.

- Then, it should be state that sensitizers (NOT PDT) can be used in diagnosis and surgery.

* ln 89: What is the meaning of higher redox state?

Answer) Thanks to your kind comment. In fact, the reduction/oxidation status of cancer cells is normally very different according to the types of cancer cells and/or origin of cancer c cells. It means that activity of reduction/oxidation is elevated in cancer cells compared to normal cells/tissues.

On the other hand, abnormal redox status of tumor microenvironment is also spotlighted in recent decades since high metabolic activity in tumor tissues promotes reduction/oxidation (redox) activities [26-30]. Many scientists have been applied abnormal redox status of tumor as a therapeutic target for stimuli-sensitive drug delivery [26-30].

- Please, replace the expression “higher redox state” by “reduction/oxidation status”.

* lns 87-89: “Compared to normal cells and tissues, tumor cells express various receptors with higher degree, higher enzymatic activity such as hyaluronidase (HAse), unbalanced immune system, poor perfusion, acidic pH environment and higher redox-status [22-24].” It is important to distinguish the tumor cells from the tumor microenvironment. Tumor cells do NOT express and unbalanced immune system.

Answer) Thanks to your kind comment. According to your comment, we revised the manuscript. To avoid confused sentences, we removed some sentences and revised.

Compared to normal cells and tissues, tumor cells are characterized by higher expression of receptors, higher enzymatic activity such as hyaluronidase (HAse), poor perfusion, acidic pH environment and higher redox-status compared to normal cells [22-24].

- It is really important to distinguish between the properties of the tumor cells and those of the tumor microenvironment.

Author Response

Answer to Reviewer 2’s second review

This is an improved version of the original manuscript. However, there are a few issues that still need to be addressed. Please, see below, after the authors’ replies to this reviewers’ original comments.

Answer) Thanks for your comment. I appreciated your comment.

The term “biodegradable” is usually used in the context of pollution. It seems to be out of the present study context and should preferentially be removed.

Answer) Thanks for your comment. According to your comment, we removed the terms of “biodegradable” from title and manuscript.

Title: Nanophotosensitizers based on Hyperbranched Chlorin e6-Hyaluronic Acid Conjugates via Disulfide Linkage for Enhanced Photodynamic Treatment of Cancer Cells

In Introduction

For present study, we designed nanophotosensitizer using hyperbranched Ce6-conjugated HA. Hyperbranched Ce6 was synthesized by conjugation of Ce6 each other using disulfide linkage.

- The wording “redox-sensitive and CD44-sensitive photodynamic therapy” is confusing.

Answer) Thanks for your comment. To avoid complexicity, CD-sensitive changed to “CD44 receptor-meadiated delivery.

The aim of this study is to synthesize novel types of nanophotosensitizers based on hyperbranched chlorin e6 (Ce6)-conjugated hyaluronic acid (HA) for CD44-receptor mediated delivery and redox-sensitive photodynamic therapy (PDT) against cancer cells.

- The authors did not address the question regarding the release of Ce6 from the nanomaterials.

Answer) Thanks for your comment. The release behavior of Ce6 from nanomaterials takes advantages from DTT, GSH and hyaluronidase. As shown in Figure 5(a) and (b), Ce6 release rate from nanomaterials was faster in the presence of 10 or 20 mM DTT (At in vitro drug release experiment, DTT was used instead of GSH) both Ce6tetraHA and Ce6 decaHA nanophotosensitizers. Furthermore, Ce6 release rate was accelerated when DTT and hyaluronidase was added (as you can see in the Figure 5(c)). Then I decided that DTT, GSH and hyaluronidase give advantages in the release behavior of Ce6 from nanophotosensitrizers

Figure 5. The effect of DTT on the Ce6 release dynamics from Ce6tetraHA (a) and Ce6decaHA (b) nanophotosensitizers. (c) The effect of HAse/DTT on the Ce6 release dynamics from Ce6tetraHA nanophotosensitizers. All values are average ± S.D. from three different experiments.

 The issue raised was: Is the experimental design adopted adequate to distinguish the emission of Ce6 from those of the conjugates and nanophotosensitizers? Please, provide emission spectra for the Ce6, as well as for Ce6tetraHA and Ce6decaHA conjugates and nanophotosensitezers at an excitation wavelength of 407 nm.

Answer) Thanks for your comment. According to your comment, we provided emission scan of Ce6 in sSupporting information.

Figure 2s shows the emission spectra of Ce6. Ce6 showed maximum peak between 650 and 700 nm as similar to Ce6tetraHA or Ce6decaHA.

Figure s2. Emission spectra of Ce6. Ce6 in DMSO was diluted with PBS (0.01M, pH 7.4) ten times for 10 % DMSO.

- Please, specify the source of the cell lines used.

- Please, specify the antibiotics added to the culture medium.

Answer) Thanks for your comment. We already specified the source of cell lines in Experimental section (2.7. Cell culture). Pratically, we purchased from Korean cell line bank.

NIH3T3 mouse fibroblast cells, U87MG human malignant glioma cells and HCT116 mouse colon carcinoma cells were obtained from Korean Cell Line Bank (Seoul, Korea). NIH3T3 cells and U87MG cells were cultured in DMEM (Gibco, Grand Island, NY, USA) supplemented with 10% (v/v) heat-inactivated fetal bovine serum (FBS) (Invitrogen) and and 1 % (v/v) penicillin-streptomycin at 37 °C in a 5 % CO2 incubator. HCT116 cells were cultured in RPMI1640 (Gibco, Grand Island, NY, USA) supplemented with 10% (v/v) heat-inactivated fetal bovine serum (FBS) (Invitrogen) and 1 % (vw/v) penicillin-streptomycin.

- If results are from eight wells, it means that they are from a single independent experiment with eight replicate wells for each condition. This must be stated in the manuscript.

Answer) Thanks for your comment. Figure 5 are results from three different experiments.

In experimental section

2.8. PDT treatment

All values are average ± S.D. (Standard deviation) from eight wells.

2.9. Intracellular Ce6 uptake of nanophotosensitizers

All values are average ± S.D. (Standard deviation) from eight wells.

2.11. ROS Generation

All values are average ± S.D. (Standard deviation) from eight wells.

Figure 6. Dark toxicity of Ce6, Ce6tetraHA and Ce6decaHA nanophotosensitizers against NIH3T3 mouse fibroblast, HCT116 human colon carcinoma and U87MG human malignant glioma cells. All values are average ± S.D. from results of a single independent experiment with eight replicate.

Figure 7. Ce6 uptake ratio (a) and fluorescence images (b) in the HCT116 cells and U87MG cells. For fluorescence images, Ce6 concentration was adjusted in 3.35 μM (2.0 μg/ml) and exposed to cells for 90 min. All values are average ± S.D. from results of a single independent experiment with eight replicate.

Figure 8. The effect of Ce6 or Ce6tetraHA and Ce6decaHA nanophotosensitizer treatment against cancer cells. ROS generation (a) and PDT treatment (b). Cells were irradiated at 664 nm (2 J/cm2). All values are average ± S.D. from results of a single independent experiment with eight replicate.

Figure 9. (a) The effect of blocking of CD44 receptor. Fluorescence images of U87MG cells. For blocking of CD44 receptor, 10 times higher amount of free HA was pre-treated for 60 min before treatment of Ce6 or nanophotosensitizers. (b) ROS generation and PDT effect on the cancer cells. For fluorescence images, Ce6 concentration was adjusted to 3.35 μM (2.0 μg/ml) and exposed to cells for 60 min. All values are average ± S.D. from results of a single independent experiment with eight replicate. *,**; p < 0.001.

- This should be clearly stated in the manuscript.

Answer) Thanks for your comment. According to your comment, we indicated it in the manuscript.

Nanophotosensitizers of Ce6tetraHA and Ce6decaHA can be instantly reconstituted in the aqueous solution such as deionized water or PBS (0.01M, pH 7.4).

Then, it should be state that sensitizers (NOT PDT) can be used in diagnosis and surgery.

Answer) Thanks for your comment. According to your comment, We indicated the state of PDT for diagnosis and surgery.

In Introduction part

Nanomedicine-based PDT also enables disease diagnosis and image-guided surgery [19-21]. Muhanna et al., reported that multimodal porphyrin lipoprotein-mimicking nanoparticle (PLPs) can be applied in positron emission tomography (PET), fluorescence imaging, and PDT [20].

Furthermore, photosensitizers are regarded as a promising candidate for diagnosis and image-guided surgery of tumors [19,20]. Photosensitizers facilitate fluorescence detection of tumor by distinguishing them from normal tissues [19]. Then, tumors can be removed by PDT or image-guided surgery using endoscope [19-21].

 Please, replace the expression “higher redox state” by “reduction/oxidation status”.

Answer) Thanks for your comment. According to your comment, we changed “higher redox state” by “reduction/oxidation status”.

Compared to normal cells and tissues, tumor cells are characterized by higher expression of receptors, higher enzymatic activity such as hyaluronidase (HAse), poor perfusion, acidic pH environment and higher reduction/oxidation (redox) status compared to normal cells [24-26].

- It is really important to distinguish between the properties of the tumor cells and those of the tumor microenvironment.

Answer) Thanks for your comment. According to your comment, distinguish between the properties of the tumor cells and those of the tumor microenvironment is important. We discussed more about it.

Compared to normal cells and tissues, tumor cells are characterized by higher expression of receptors, higher enzymatic activity such as hyaluronidase (HAse), poor perfusion, acidic pH environment and higher reduction/oxidation (redox) status compared to normal cells [24-26]. Then, tumor microenvironment has quite different physiological states compared to normal tissues.

Reviewer 3 Report

I am satisfied with most of the answers of the authors, and I believe the manuscript has been improved. However, the assignment of NMR signals only demonstrates that chlorin e6 has been functionalized, but does not prove that the proposed structures, tetramer and decamer, were formed. Therefore, the conjugates that are described are based on the assumptions that the reactions occurred as desired. As these are complex structures, I suggest that mass spectra should be provided (Ideally HPLC-MS).

Please be attention to the synthesis schemes. The structure of Ce6 is incorrect.

Author Response

I am satisfied with most of the answers of the authors, and I believe the manuscript has been improved. However, the assignment of NMR signals only demonstrates that chlorin e6 has been functionalized, but does not prove that the proposed structures, tetramer and decamer, were formed. Therefore, the conjugates that are described are based on the assumptions that the reactions occurred as desired. As these are complex structures, I suggest that mass spectra should be provided (Ideally HPLC-MS).

Answer) Thanks for your comment. At this moment, we couldn’t get HPLC-MS data. However, we added additional proton NMR results in supporting information previously and indicated drug contents in Table 1. More precise characterization of hyperbranched Ce6 will be reported other repot in the near future.

Please be attention to the synthesis schemes. The structure of Ce6 is incorrect.

Answer) Thanks for your comment. I appreciated your comment. According to your comment, we fully revised the manuscript and corrected the chemical structure of Ce6.

Round 3

Reviewer 3 Report

As the authors could not provide additional spectroscopic data, I’m concerned about the unsupported assumptions regarding the structures claimed in this work. I would like to see the authors address this critical issue and, therefore, I suggest the authors revise and resubmit the paper.

Author Response

Answer to reviewer’s comment

As the authors could not provide additional spectroscopic data, I’m concerned about the unsupported assumptions regarding the structures claimed in this work. I would like to see the authors address this critical issue and, therefore, I suggest the authors revise and resubmit the paper.

Answer) Thanks for your comment. According to your comment, we additionally synthesized Ce6tetra-MePEG 2K and Ce6deca-MePEG 5K conjugates, and we estimated the number of Ce6 in Ce6 tetramer and Ce6 decamer. The Ce6 number in Ce6 tetramer and Ce6 decamer was estimated as 3.64 and 9.1, respectively. Figures and Table were added to supporting information as (Figure s2 and Table s1). Furthermore, we added fluorescence spectroscopic results of Ce6, Ce6 tetramer and Ce6 decamer. Thanks.

___________________________________________________________

In main text

The estimated number of Ce6 of Ce6 tetramer and decamer was 3.64 and 9.1, respectively (Table s1).

In Supporting information

Materials

 Methoxy poly(ethylene glycol)-amine (MePEG-amine) with molecular weight of 2,000 g/mol and 5,000 g/mol (MePEG amine 2K, MePEG-amine 5K) was purchased from Sunbio Co. Ltd. (Seoul, Korea).

Ce6tetra-MePEG 2K conjugates

  27.5 mg of Ce6 tetramer was dissolved in 10 ml DMSO with EDAC (1.91 mg, 0.01 mM) and NHS (1.15 mg, 0.01 mM). This solution was magnetically stirred for 9h and then mixed with 20 mg MePEG-amine 2K in 5 ml DMSO. This solution was stirred for 36 h and then put into dialysis membrane (MWCO: 8,000 g/mol) to dialyze against water over 2 days. Following this, resulting solution was lyophilized for 3 days to obtain Ce6tetra-MePEG 2K conjugates as a solid. This solid was stored in refrigerator at -20 oC. Yield was approximately 98.6% (w/w) from mass measurement: Yield = [Weight of final product/(Weight of Ce6 tetramer + weight of MePEG-amine 2K)] × 100.

Ce6deca-MePEG 5K conjugates

Ce6 decamer (70 mg) was dissolved in 15 ml DMSO with EDAC (1.91 mg, 0.01 mM) and NHS (1.15 mg, 0.01 mM). This solution was stirred magnetically for 9h and then mixed with 50 mg MePEG amine (M.W. = 5,000 g/mol) in 5 ml DMSO. This solution was further stirred for 36h and then put into dialysis membrane (MWCO: 8,000 g/mol). This was dialyzed against water over 2 days and lyophilized for 3 days to obtain Ce6deca-MePEG 5K conjugates as a solid. Final product was used to analyse or stored in refrigerator at -20 oC. Yield was approximately 99.2 % (w/w) from mass measurement: Yield = [Weight of final product/(Weight of Ce6 decamer + weight of MePEG amine 5K)] × 100.

Results

Furthermore, Ce6 tetramer and Ce6 decamer was conjugated with MePEG-amine  2K and MePEG-amine 5K to characterize the hyperbranched Ce6 as shown in Figure s2. As shown in Figure s2(a) and (b), ethylene proton of MePEG and methylene proton of Ce6 tetramer or Ce6 decamer was confirmed at 3.6 ppm and 1.6 ppm, respectively, and the number of Ce6 in hyperbranched Ce6 was estimated from these peaks. As abbreviated in Table s1, the estimated number of Ce6 in hyperbranched Ce6 was 3.64 and 9.1, respectively. Even though the experimental number of Ce6 in Ce6 tetramer and Ce6 decamer was slightly lower than theoretical value, Ce6 tetramer and Ce6 decamer was successfully synthesized.

Figure s3 shows the emission spectra of Ce6. Ce6 showed maximum peak between 650 and 700 nm as similar to Ce6tetraHA or Ce6decaHA. Furthermore, the fluorescence spectra of Ce6 tetramer and Ce6 decamer were compared with Ce6 as shown in Figure s4. As shown in Figure s4, the fluorescence spectra of Ce6 tetramer and Ce6 decamer were not significantly different to the fluorescence spectra of Ce6.

Figure s2. 1H NMR spectra of Ce6 tetramer–MePEG 2K conjugates (a) and Ce6 decamer-MePEG 5K conjugates (b). For measurement of 1H NMR (500 mHz) spectra, Ce6 tetramer–MePEG 2K conjugates and Ce6 decamer-MePEG 5K conjugates were dissolved in deuterated DMSO (DMSO d6).

Figure s4. Fluorescence spectra of Ce6, Ce6 tetramer and Ce6 decamer. Ce6, Ce6 tetramer and Ce6 decamer was dissolved in DMSO (concentration: 0.1 mg/ml).
